# SCALELONG: A MULTI-TIMESCALE BENCHMARK FOR LONG VIDEO UNDERSTANDING

**David Ma**[1,*] **Huaqing Yuan**[1,*] **Xingjian Wang**[1,*] **Qianbo Zang**[1,*] **Tianci Liu**[1], **Xinyang He**[1]
**Yanbin Wei**[1], **Jiawei Guo**[1], **Jiahui Ni**[1], **Zhenzhu Yang**[1], **Meng Cao**[2], **Shanghaoran Quan**[1]
**Yizhi LI**[1], **Wangchunshu Zhou**[1,3] **Jiaheng Liu**[1,4], **Wenhao Huang**[1], **Ge Zhang**[1,†] **Shiwen Ni**[5,6,†]
**Xiaojie Jin**[1,†]
[1]M-A-P   [2]MBZUAI   [3]OPPO   [4]NJU   [5]AIRI,SUAT   [6]SIAT,CAS

## ABSTRACT

Although long-video understanding demands that models capture hierarchical temporal information—from clip and shot to event and story—existing benchmarks either neglect this multi-scale design or scatter scale-specific questions across different videos, preventing direct comparison of model performance across timescales on the same content. To address this, we introduce Scale-Long, the first benchmark to disentangle these factors by embedding questions targeting four hierarchical timescales—clip, shot, event, and story—all within the same video content. This within-content multi-timescale questioning design enables direct comparison of model performance across timescales on identical videos. ScaleLong features 269 long videos (avg. 86 min) from 5 main categories and 36 sub-categories, with 4–8 carefully designed questions, with at least one question targeting each timescale. Evaluating 23 MLLMs reveals a distinct U-shaped performance trend: higher accuracy at the shortest (clip) and longest (story) timescales, with a dip at intermediate levels. Furthermore, ablation studies demonstrate that increased visual token capacity consistently enhances reasoning across all timescales. ScaleLong offers a crucial fine-grained, multi-timescale benchmark for advancing MLLM capabilities in long-video understanding. The code and dataset are available at `https://github.com/multimodal-art-projection/ScaleLong`.

## 1 INTRODUCTION

Recent advances in Multimodal Large Language Models (MLLMs) have significantly enhanced their ability to integrate and interpret complex inputs such as text, images, and videos (Liu et al., 2023a; Chen et al., 2024b; Guo et al., 2024; Zhang et al., 2024; Zhu et al., 2025; Team et al., 2025). Consequently, a variety of benchmarks have been developed to gauge their video understanding capabilities across different scopes and tasks (Wu et al., 2024; Wang et al., 2023; Fu et al., 2024a; Li et al., 2024e; Ma et al., 2025). True comprehension of a long video, much like human understanding, requires seamlessly integrating observations across a hierarchy of temporal scales: from recognizing a momentary gesture to grasping the overarching plot.

However, current long-video benchmarks are ill-equipped to assess the multi-timescale capabilities of MLLMs—specifically, their distinct abilities across varying temporal granularities. By typically using isolated short segments (Li et al., 2024d) or evaluating different temporal scales across entirely different videos (Zhou et al., 2024), these benchmarks inherently conflate temporal granularity with content variability. This makes it exceedingly difficult to disentangle a MLLM's true performance at each specific timescale from content-driven adaptations. Thus, a rigorous, fine-grained methodology is critically needed to evaluate how MLLMs apply these distinct temporal capabilities to understand the hierarchical temporal structures within long videos.

---

[*]Equal contribution.
[†]Corresponding authors.

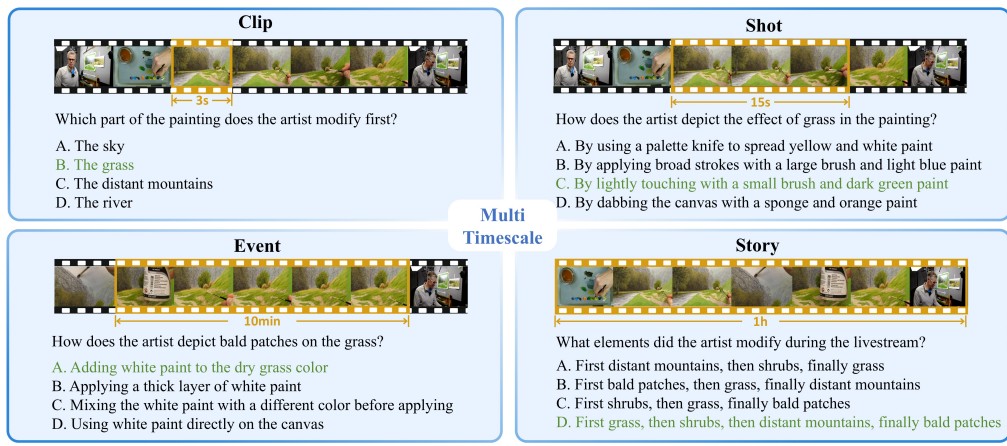

Figure 1: Representative samples from ScaleLong. Each sample in ScaleLong comprises a video paired with carefully designed questions, structured across four hierarchical temporal scales. The correct answers are indicated in green.

To bridge this gap, we introduce ScaleLong, a benchmark tailored for the fine-grained evaluation of MLLMs' multi-timescale capabilities in long videos. Its core feature is embedding questions targeting four hierarchical temporal scales (Clip, Shot, Event, Story) all within the same video content. This 'within-content' design enables direct comparison of an MLLM's performance across these distinct temporal granularities on identical narratives, thereby isolating its abilities at each specific scale. ScaleLong includes 269 videos (averaging 86 minutes), each annotated with 4–8 questions, ensuring at least one question per time scale. As illustrated in Fig. 2, the benchmark spans 5 major categories and 36 subcategories, enabling a comprehensive evaluation of MLLMs' understanding of long videos across diverse timescales.

Leveraging ScaleLong, extensive evaluations of 23 MLLMs—encompassing 19 open-source and 4 proprietary models—consistently reveal a U-shaped performance curve across the defined temporal scales. These models generally perform better on questions at the shortest (Clip) and longest (Story) temporal scales, while performance noticeably drops at intermediate levels (Shot and Event). This pattern suggests that current MLLMs often excel at processing either highly localized visual details or overarching narrative structures, yet face challenges with intermediate temporal contexts. Furthermore, targeted experiments conducted on ScaleLong indicate that an increased allocation of visual tokens systematically enhances MLLMs' performance across all evaluated timescales, providing valuable insights for future advancements in model development.

In summary, our work makes three primary contributions:

- **ScaleLong:** We introduce ScaleLong, specifically designed to assess the multi-timescale capabilities of MLLMs in long videos. By embedding questions at four hierarchical temporal scales (Clip, Shot, Event, and Story) within the same video content, it enables rigorous evaluation of MLLMs at each distinct scale. ScaleLong includes 269 diverse long videos (averaging 86 minutes), with 4-8 questions per video (at last one per scale), across 5 major categories and 36 subcategories.

- **Comprehensive MLLM Evaluation and Insights**: Our extensive evaluation of 23 MLLMs on ScaleLong reveals a consistent U-shaped performance trend. MLLMs generally exhibit stronger comprehension at the shortest (Clip) and longest (Story) temporal scales, while their performance discernibly dips at intermediate scales (Shot and Event). This finding offers critical insights into how MLLMs process information at distinct temporal granularities in long videos.

- **Insights for MLLM Development**: Evaluations on ScaleLong provide key insights for MLLM enhancement. For instance, increasing visual token allocation consistently enhances performance across all evaluated timescales. Furthermore, analysis of model error patterns reveals persistent weaknesses, offering guidance for future model improvements in long-video understanding.

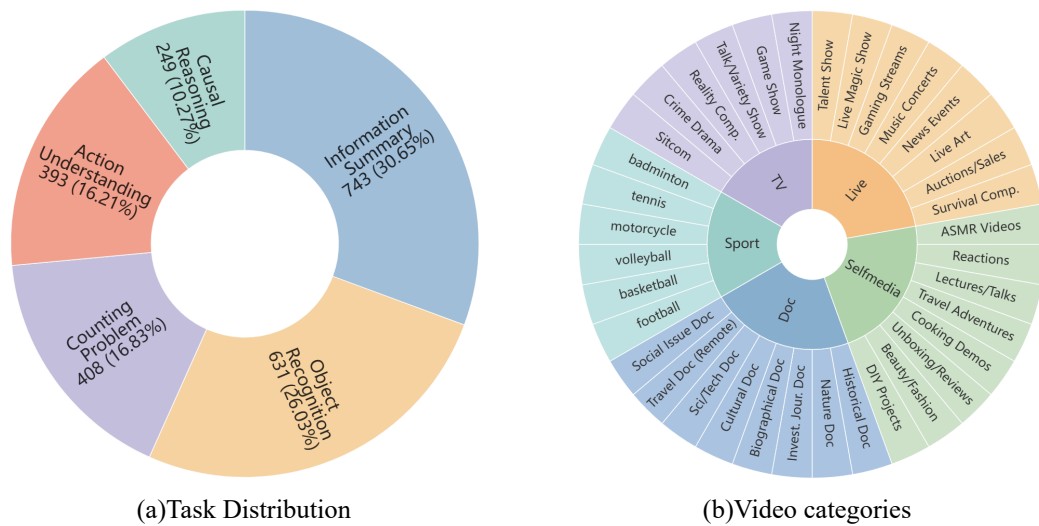

(a)Task Distribution                 (b)Video categories

Figure 2: (a) Task distribution in ScaleLong. ScaleLong consists of a total of 5 tasks, ensuring comprehensive evaluation of the model's capabilities. (b) Video Categories. ScaleLong includes videos spanning 5 major categories and 36 subcategories, ensuring diverse topical coverage.

## 2 SCALELONG

### 2.1 OVERVIEW

ScaleLong is specifically engineered for the fine-grained assessment of the multi-timescale capabilities of MLLMs in long videos. The benchmark comprises 269 diverse videos, each averaging 86 minutes and annotated with 4-8 questions. As illustrated in Fig. 1, these questions address four hierarchical temporal scales: Clip, Shot, Event, and Story. Key features of ScaleLong include:

**Multi Timescale Queries**: Unlike existing benchmarks, ScaleLong structures queries at four meticulously defined temporal scales—Clip, Shot, Event, and Story—all within each individual video. This 'within-content' embedding of questions targeting multiple, distinct temporal scales is crucial: it effectively decouples the assessment of temporal understanding of specific video content. Such a design enables precise evaluation of how MLLMs handle different temporal granularities while keeping the narrative context consistent.

**Diverse Video Content and Task Design**: For comprehensive MLLM evaluation, ScaleLong offers extensive content diversity, featuring 5 main video categories (e.g., Sports, Documentaries) spanning 36 subcategories. It also incorporates 5 distinct task types (e.g., Causal Reasoning, Action Understanding) designed to probe deeper comprehension. This structured variety ensures representative assessment across diverse, real-world long-video scenarios.

### 2.2 MULTI-TIMESCALE HIERARCHIES

The Multi-Timescale Hierarchies within ScaleLong are established by categorizing questions into four distinct temporal levels. This classification is based on the video duration essential for answering each question and how relevant information is distributed across the frames. These levels are detailed as follows:

**Clip**: Questions solvable by analyzing a few consecutive frames, spanning only a few seconds (e.g., up to 3seconds), typically involving recognizing instantaneous actions, immediate visual details, or straightforward objects.

**Shot**: Questions requiring the analysis of information from multiple frames within a single continuous shot, typically ranging from 4 to 15 seconds. These questions require interpreting short-term dynamics, simple actions, character interactions, or semantic coherence within this timeframe.

**Event**: Questions concerning significant events that span multiple consecutive shots, with durations from 16 seconds up to 10 minutes. These questions require integrating information across scenes, interpreting event sequences, identifying contextually relevant frames, and understanding more complex narrative developments or causal links.

**Story**: Questions addressing content from the entire video or substantial portions thereof, typically exceeding 10 minutes. These require holistic comprehension of the overall narrative, including overall narrative logic, causal relationships, character development, thematic analysis, or long-term dependencies across the video.

### 2.3 TASK TYPES

ScaleLong incorporates five distinct task types, each designed to rigorously evaluate different facets of an MLLM's comprehension abilities:

**Causal Reasoning (CR)**: Questions requiring inference about causal relationships within the video content. These tasks assess the model's ability to deduce cause-and-effect dynamics and logical connections.

**Object Recognition (OR)**: Questions involving the identification and distinction of specific objects, scenes, or their attributes within the video. These tasks evaluate visual perception and fine-grained recognition capabilities.

**Action Understanding (AU)**: Questions focused on interpreting the actions, movements, or behaviors of entities (e.g., characters, objects) in the video. These tasks assess the capacity to comprehend dynamic interactions and temporal movements.

**Information Summary (IS)**: Questions that require summarizing or generalizing main content, key points, or details from the video. These tasks evaluate the ability to synthesize information and extract essential concepts.

**Counting Problems (CP)**: Questions that are designed to rigorously assess an MLLM's capacity for precise numerical grounding—that is, its ability to accurately identify, track, and count instances based purely on video content.

### 2.4 ANNOTATION METHODOLOGY AND QUALITY CONTROL

#### 2.4.1 ANNOTATION METHODOLOGY

ScaleLong ensures high-quality annotations through a multi-phase process involving curated video selection, structured question design, and multi-round quality control. This process emphasizes content-based understanding, requiring questions to target video-specific information, answers to be thoroughly video-grounded, and dependencies on absolute time cues or external knowledge to be eliminated.

**Video Curation and Collection**: Our video acquisition process begins by defining 5 principal categories (further detailed into 36 subcategories) to cover diverse real-world scenarios. YouTube videos, typically around one hour in length, are manually sourced for these categories. Each selected video undergoes inspection for high visual clarity, substantial information density, and appropriate duration, resulting in a final corpus of 269 videos.

**Question, Answer, and Distractor Generation**: For each video, annotators first conduct a full viewing. They then design 8 questions (two for each of the four defined temporal hierarchy levels), ensuring a balance in task types. Correct answers are derived from careful analysis of multimodal information within the video. Each question is accompanied by one correct answer and three plausible distractors, which are constructed based on ten predefined types to offer varied challenges and facilitate error analysis.

#### 2.4.2 RIGOROUS QUALITY CONTROL

Our quality control protocol involves two principal rounds with distinct objectives:

Table 1: Comparison with other benchmarks, where the abbreviations are defined as follows: **Anno.** (Annotation Method), **A** (Automatic Annotation), **M** (Manual Annotation), **#Genres** (Number of Video Genres). MTS is the abbreviation for Multi-Timescale, and IV-MTS is the abbreviation for Intra-Video Multi-Timescale.

| Benchmark | #Videos | Duration. (s) | #Tasks | #QA Pairs | Anno. | #Genres | MTS | IV-MTS |
|---|---|---|---|---|---|---|---|---|
| MSVD-QA | 1,970 | 10 | - | 13,157 | A | - | ✗ | ✗ |
| MSRVTT-QA | 2,900 | 15 | - | 72,821 | A | - | ✗ | ✗ |
| ActivityNet-QA | 5,800 | 111 | 4 | 800 | M | 8 | ✗ | ✗ |
| NExTQA | 1,000 | 44 | 4 | 8,564 | M | - | ✗ | ✗ |
| MVBench | 3,641 | 16 | 20 | 4,000 | A | - | ✗ | ✗ |
| CinePile | 9,396 | 160 | 5 | 303,828 | M & A | 1 | ✗ | ✗ |
| EgoSchema | 5,063 | 180 | - | 5,063 | M & A | - | ✗ | ✗ |
| LVBench | 103 | 4,101 | 6 | 1,549 | M | 21 | ✗ | ✗ |
| LONGVIDEOBENCH | 3,763 | 473 | 17 | 6,678 | M | 10 | ✗ | ✗ |
| HourVideo | 500 | 2,742 | 4 | 12,976 | M & A | - | ✗ | ✗ |
| ALLVB | 1,376 | 7,200 | 9 | 252,000 | A | 16 | ✗ | ✗ |
| Video-MME | 900 | 1,024 | 12 | 2,700 | M | 30 | ✓ | ✗ |
| MoVQA | 100 | 992 | 6 | 21,953 | M | 1 | ✓ | ✗ |
| MLVU | 1,730 | 930 | 9 | 3,102 | M | 31 | ✓ | ✗ |
| **ScaleLong** | 269 | 5,160 | 5 | 1747 | M | 36 | ✓ | ✓ |

**First-Round Quality Control**: This round focuses on the foundational correctness, clarity, and consistency of all annotations. Question stems are verified for precision. Critically, absolute time localizations are replaced with descriptive cues to compel content-based reasoning rather than timestamp reliance. Answer options are thoroughly checked: correct answers must be unambiguously video-grounded, and distractors plausible yet definitively incorrect. Annotations also undergo checks to prevent an undue concentration of questions within limited segments of the video's timeline, and to validate all categorizations (e.g., temporal levels, task types, distractor types).

**Second-Round Quality Control**: This round focuses on eliminating confounding factors and ensuring questions exclusively assess understanding derived from the video content. Questions solvable through common world knowledge or reliant on external prior information, rather than video-specific details, are systematically revised or removed to nullify such external dependencies. Finally, to uphold dataset integrity, any questions exhibiting persistent ambiguities (e.g., unclear grounding, indistinct features, or problematic categorization) are rigorously discarded.

## 3 COMPARISON WITH OTHER VIDEO BENCHMARKS

Existing video understanding benchmarks, as detailed in Table 1, are broadly categorized into short-video and long-video formats. Short-video benchmarks like NExTQA (Xiao et al., 2021) and MVBench (Li et al., 2024b) utilize sub-minute clips, which restrict their capacity for evaluating long-range temporal understanding. Recent long-video benchmarks—including CinePile (Rawal et al., 2024), EgoSchema (Mangalam et al., 2023), MoVQA (Zhang et al., 2023b), MLVU (Zhou et al., 2024), Video-MME (Fu et al., 2024b), LongVideoBench (Wu et al., 2024), and ALLVB (Tan et al., 2025)—feature extended durations. However, they generally do not decouple the targeted temporal scales of questions from specific video content. This inherent coupling hinders a precise assessment of how MLLMs handle varying temporal granularities and, consequently, their distinct multi-timescale capabilities.

ScaleLong adopts a **'quality over quantity'** philosophy, designed as a high-fidelity **evaluation benchmark**, not a training corpus. While large-scale, model-synthesized benchmarks like ALLVB risk introducing biases and quality fluctuations, our focus on meticulous manual annotation ensures high data integrity. We believe this compact, high-quality approach, proven effective by benchmarks like GPQA (Rein et al., 2024), offers a more trustworthy and statistically sound foundation for MLLM assessment.

Unlike most existing long-video benchmarks which, as noted, typically conflate temporal scale assessment with disparate video content, ScaleLong is distinguished by its primary design attribute: the Intra-Video Multi-Timescale nature. This principle dictates that within every long video, ques-

Table 2: The performance of proprietary and open-source MLLMs on ScaleLong across granularities and task types. For each timescale and task, the best performance is indicated in bold, and the second-best performance is indicated with underlining.

| Models | Date | Input | Granularities | | | | Task Types | | | | | Overall |
|---|---|---|---|---|---|---|---|---|---|---|---|---|
| | | | Clip | Shot | Event | Story | CR | OR | AU | IS | CP | |
| Human | N/A | Whole Video | 92.8 | 91.3 | 88.9 | 91.0 | 90.7 | 91.2 | 89.3 | 92.7 | 89.8 | 91.0 |
| *Proprietary Models* | | | | | | | | | | | | |
| Gemini 2.0 Flash | 2025-02 | 256 frm | 65.7 | 52.4 | 48.4 | 53.4 | 53.5 | 64.8 | 54.6 | 55.9 | 41.9 | 55.0 |
| GPT-4o | 2024-05 | 64 frm | 61.8 | 50.7 | 51.0 | 58.0 | 58.3 | 62.6 | 57.4 | 60.1 | 36.0 | 55.4 |
| Doubao 1.5-VL Pro | 2025-01 | 256 frm | 66.4 | 52.8 | 55.2 | 60.2 | 57.1 | 67.0 | 55.1 | 64.5 | 43.3 | 58.7 |
| Gemini 2.5 Pro | 2025-03 | 256 frm | **71.5** | **62.8** | **68.0** | **69.0** | **66.0** | **72.5** | **65.8** | **74.6** | **51.2** | **67.9** |
| *Open source Models* | | | | | | | | | | | | |
| LLaVA-Mini | 2025-01 | 256 frm | 29.7 | 25.3 | 28.8 | 25.2 | 27.6 | 29.8 | 29.4 | 27.2 | 22.1 | 27.3 |
| LongVILA | 2024-08 | 32 frm | 29.1 | 28.3 | 23.8 | 28.6 | 28.0 | 29.2 | 30.0 | 26.8 | 23.9 | 27.5 |
| LongVU | 2024-10 | 32 frm | 40.9 | 37.2 | 33.5 | 35.6 | 43.9 | 44.1 | 38.1 | 21.7 | 36.8 | 36.8 |
| Phi-3.5 | 2024-04 | 64 frm | 44.8 | 35.8 | 34.3 | 43.0 | 43.3 | 47.9 | 33.9 | 40.1 | 30.2 | 39.5 |
| LongVA | 2024-06 | 256 frm | 50.3 | 40.2 | 38.8 | 43.8 | 49.3 | 53.1 | 38.2 | 45.4 | 28.8 | 43.3 |
| Flash-VStream | 2024-06 | 256 frm | 46.9 | 42.7 | 39.8 | 47.0 | 48.7 | 48.9 | 39.0 | 34.1 | 48.6 | 44.1 |
| Phi-4 | 2025-03 | 128 frm | 50.0 | 42.9 | 42.1 | 45.5 | 50.3 | 53.9 | 44.0 | 46.2 | 30.8 | 45.2 |
| LLaVA-OV-7B(SI) | 2024-08 | 128 frm | 50.8 | 39.5 | 44.4 | 47.3 | 41.4 | 52.2 | 44.1 | 49.3 | 34.3 | 45.5 |
| MiniCPM-V | 2024-08 | 64 frm | 51.0 | 42.9 | 43.1 | 47.8 | 49.0 | 55.2 | 42.9 | 48.5 | 32.5 | 46.2 |
| Qwen2-VL-7B | 2024-09 | 8 frm | 51.6 | 45.9 | 46.5 | 48.3 | 51.6 | 51.9 | 53.1 | 50.6 | 33.9 | 48.1 |
| LLaVA-Video-7B | 2024-10 | 32 frm | 57.9 | 46.8 | 50.2 | 48.0 | 47.1 | 55.4 | 52.3 | 54.4 | 39.9 | 50.8 |
| InternVL2-5-8B | 2024-12 | 128 frm | 60.2 | 48.5 | 48.5 | 52.3 | 52.0 | 61.5 | 47.4 | 50.8 | 39.3 | 50.9 |
| Qwen2.5-VL-7B | 2025-01 | 256 frm | 52.8 | 50.0 | 49.5 | 52.7 | 50.2 | 54.5 | 53.3 | 55.3 | 39.9 | 51.2 |
| Aria | 2024-10 | 256 frm | 57.2 | 46.5 | 48.7 | 53.4 | 49.3 | 60.5 | 48.5 | 52.4 | 41.6 | 51.5 |
| LLaVA-OV-72B | 2024-08 | 64 frm | 56.1 | 49.5 | 51.5 | 55.2 | 53.6 | 59.3 | 53.5 | 52.3 | 45.5 | 53.1 |
| InternVL2-5-26B | 2024-12 | 128 frm | 60.2 | 50.1 | 48.5 | 56.8 | 57.9 | 61.7 | 53.6 | 52.9 | 43.6 | 53.9 |
| LLaVA-Video-72B | 2024-10 | 128 frm | 60.0 | 50.7 | 53.2 | 51.8 | 54.6 | 62.7 | 55.5 | 55.2 | 39.0 | 53.9 |
| InternVL2-5-38B | 2024-12 | 256 frm | 61.8 | 53.5 | 54.1 | 55.5 | 53.9 | 65.4 | 58.8 | 58.1 | 40.7 | 56.3 |
| InternVL2-5-78B | 2024-12 | 128 frm | 65.2 | 54.3 | 53.4 | 61.5 | 57.2 | 65.4 | 54.4 | 63.9 | 46.2 | 58.6 |

tions are specifically designed to target multiple, distinct temporal scales. This inherent characteristic—materialized through questions probing four hierarchical levels (*Clip*, *Shot*, *Event*, and *Story*) all within the same video narrative—is fundamental to the engineering of ScaleLong for the precise assessment of multi-timescale capabilities of MLLMs in Long Videos. Such a design feature directly facilitates a disentangled evaluation; the distinct capabilities of an MLLM at various temporal granularities are thereby measured against the same video content, allowing for a clear separation of temporal understanding performance from content-specific reactions.

## 4 EXPERIMENTS

In this section, we evaluate representative MLLMs on ScaleLong—first outlining the setting of experiments, then analyzing the results of the 23 MLLMs on ScaleLong. We assess how the visual tokens shape long video understanding, and finally examine error rates by distractor type to identify key failure modes.

### 4.1 SETTINGS

We evaluate a total of 23 MLLMs, comprising 4 leading commercial models—Gemini-2.5-pro (DeepMind, 2025), Gemini-2.0-flash (Team et al., 2024), GPT-4o (OpenAi, 2024) and Doubao-1.5-vision-pro (Doubao Team, 2025)—and 19 open-source models spanning from 7 billion to 78 billion parameters, including representative models such as Qwen2.5-VL (Bai et al., 2025), InternVL2.5 (Chen et al., 2024a) and LLaVA-OneVision (Liu et al., 2023b).

### 4.2 MAIN RESULTS

Table 2 presents the performance of all evaluated models across four timescale questions as well as five task types defined in ScaleLong. In this subsection, for all experiments, we fix the resolution at 240p and use the highest frame count we have tested. We draw the following key observations:

**Model performance varies significantly across timescales:** In long-video understanding, models exhibit a pronounced U-shaped trend across four timescales—Clip, Shot, Event, and Story. Accuracy peaks at the extremes (Clip and Story) but dips markedly at intermediate timescales (Shot and Event). This suggests MLLMs excel at capturing brief cues and overarching narratives but struggle with temporal coherence over moderate-length segments. For instance, Gemini 2.5 Pro scores 71.5% on Clip and 69.0% on Story, yet drops to 62.8% on Shot and 68.0% on Event. This U-shaped pattern is consistent across all models. Crucially, this MLLM performance fluctuation contrasts sharply with human consistency. Human evaluators demonstrate near-uniform accuracy across timescales (e.g., 92.8% on Clip, 91.3% on Shot, 88.9% on Event, 91.0% on Story), confirming consistent question difficulty. Thus, the significant model performance variations reflect their inherent limitations in handling different temporal spans, not uneven question design.

**Performance differences across models are notable:** Closed-source models consistently outperform open-source ones across all timescales. For example, Gemini 2.5 Pro, the leading closed-source model, surpasses the best open-source counterpart (InternVL2.5-78B) by at least 6.3 percentage points on Clip and 14.6 points on Event. Within the InternVL2.5 series, scaling from 8B to 78B yields steady accuracy gains across timescales. Despite these improvements, even the top models remain substantially below human performance. Gemini 2.5 Pro (67.9% overall) trails the human baseline (91.0% overall) by 23.1 percentage points, with the widest disparity (40.6 points) seen on Shot-scale tasks (human 91.3% vs. GPT-4o 50.7%). This underscores the considerable room for improvement in MLLMs.

**MLLMs exhibit substantial performance disparities across task types:** For most models, Objective Recognition tasks achieve the highest accuracy, while Counting Problems tasks incur the lowest. For example, Doubao 1.5-VL Pro shows a 23.7 percentage-point gap between OR and CP, and GPT-4o exhibits a 26.6-point difference. This consistent gap highlights that MLLMs' ability to perform counting remains a critical area for improvement in long-video understanding. In stark contrast, human performance remains consistently high across all task types, indicating that the observed MLLM disparities reflect genuine weaknesses in specific reasoning skills.

In summary, our findings pinpoint two critical directions for future MLLM development. The U-shaped performance across timescales reveals a core deficit in modeling intermediate temporal granularities, demanding architectures that better handle mid-range coherence. Furthermore, the pronounced weakness in specific tasks like counting underscores the need for targeted training to improve these reasoning skills. Addressing these dual challenges is essential for closing the substantial gap with human performance.

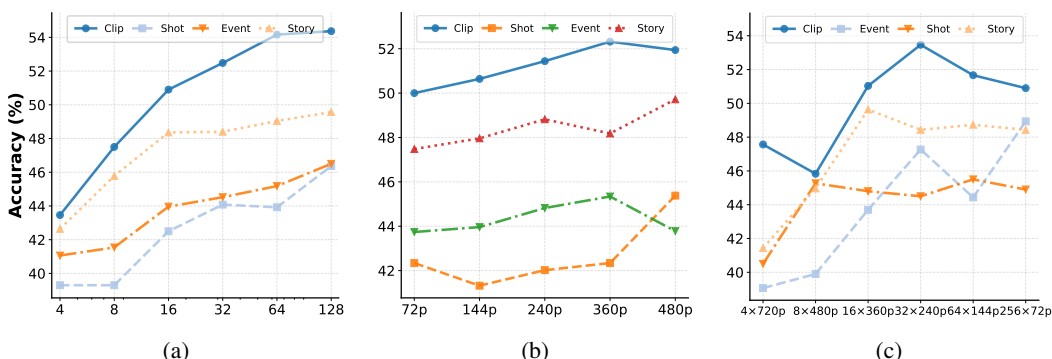

Figure 3: Comparison of model performance under: (a) varying frame counts, (b) varying video resolutions, and (c) different frame–resolution combinations.

## 4.3 ABLATION STUDY

*To investigate how total visual-token count and its allocation between frame number and resolution affect MLLM performance in multi-timescale long-video understanding*, we conduct two ablation studies:

**Scaling Effect.** How does performance change as we increase the total number of visual tokens—either by sampling more frames or by raising resolution?

**Token Allocation.** When the total visual-token budget is held constant, does distributing tokens across more frames or into higher resolution yield greater gains?

### 4.3.1 ISOLATED SCALING OF FRAME NUMBER AND RESOLUTION

In this subsection, we evaluate how allocating extra visual tokens—temporally by sampling more frames or spatially by increasing resolution—affects model performance, as illustrated in Fig. 3.

**Under a fixed resolution, increasing input frames consistently improves performance, with the greatest gains on Clip-level tasks.** As shown in Fig. 3(a), all models benefit from more frames, but the impact varies by timescale. Clip-level tasks, which depend on fine-grained temporal changes, see the most substantial improvement. In contrast, gains are more moderate for Shot and Event levels. Notably, Event-level accuracy peaks at 64 frames, suggesting that too many frames can introduce redundancy. For Story-level tasks, performance gains are limited, indicating that a smaller number of frames is often sufficient for long-range reasoning.

**Under a fixed frame count, raising resolution offers moderate gains but is less effective than adding frames and can have negative returns.** As seen in Fig. 3(b), higher resolution generally helps across all timescales, but the accuracy uplift is considerably smaller than that achieved by increasing the frame count. Furthermore, the trend is not strictly monotonic; performance on Clip-level tasks slightly dips at the highest resolution (480p), suggesting that excessive spatial detail can introduce noise that hinders short-span reasoning.

### 4.3.2 TOKEN ALLOCATION: FRAMES VS. RESOLUTION

To analyze the trade-off between temporal and spatial information under a fixed visual-token budget, we evaluated several frame-resolution combinations, averaging results across three Qwen2.5-VL models (e.g., Qwen2.5-VL-7B, Qwen2.5-VL-32B, Qwen2.5-VL-72B).

**The optimal allocation of visual tokens is highly dependent on the target timescale.** As shown in Fig. 3(c), short-span **Clip**-level tasks benefit most from high temporal density (many low-res frames). Conversely, long-range **Story**-level understanding peaks with a more balanced setting, showing diminishing returns from excessive frame counts. Intermediate timescales like **Shot** and **Event** also thrive on a moderate balance, as performance degrades when either temporal detail or spatial clarity is overly sacrificed. This highlights that no single configuration is universally optimal for long-video understanding.

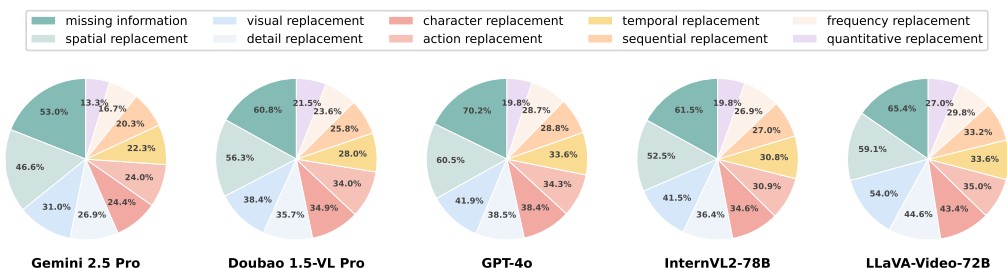

Figure 4: Distractor-specific error rate distribution across five MLLMs.

### 4.4 ERROR RATES ACROSS DIFFERENT DISTRACTOR TYPES

To analyze error patterns in long-video understanding, we evaluate several MLLMs on the ten distractor types in ScaleLong, as shown in Fig. 4. Although overall error rates are comparable across models, two categories—missing information and spatial replacement—stand out with the highest failure rates. For example, Gemini 2.5 Pro, our best-performing model, erroneously accepts missing-information distractors 53% of the time and spatial-replacement distractors 46.6% of the

time. These findings indicate a pervasive insensitivity to the completeness of evidential support, as well as a notable deficiency in reasoning about spatial relationships within complex video sequences.

In contrast, models exhibit markedly stronger performance on frequency misdirection and quantitative misdirection. GPT-4o misclassifies these distractors only 19.8% and 28.7% of the time, respectively, while Gemini 2.5 Pro's error rates are even lower (13.3% and 16.7%). This suggests that, despite their struggles with semantic completeness and spatial inference, current MLLMs are adept at leveraging statistical and numerical cues. Together, these results highlight the need for future work to incorporate mechanisms—either through architectural enhancements or targeted training curricula—that explicitly verify evidential completeness and model multi-view spatial configurations in long-video contexts.

## 5    RELATED WORK

**Multimodal Large Language Models**   Multimodal LLMs (MLLMs), which pair visual encoders with large language models, excel at image and short-video tasks (e.g., LLaVA-onevision (Li et al., 2024a), Otter (Li et al., 2023), mPLUG-Owl (Ye et al., 2023)). To address the challenges of long-video understanding, recent MLLMs introduce specialized designs—such as using a ViT + Q-Former in Video-LLaMA (Zhang et al., 2023a), efficient visual compression in LLaMA-Vid (Li et al., 2024c), scalable multi-event modeling in mPLUG-owl3 (Ye et al., 2024), and audio integration in LLaVA-Octopus (Zhao et al., 2025). In parallel, other approaches leverage massively expanded multimodal pretraining to improve temporal and contextual reasoning, as seen in state-of-the-art models like InternVL2.5 (Chen et al., 2024a) and Qwen2.5-VL (Bai et al., 2025).

**Video Understanding Benchmarks**   Video understanding benchmarks have evolved significantly, progressing from short-clip formats (e.g., MVBench (Li et al., 2024b), NExT-QA (Xiao et al., 2021)), through mid-length tasks (e.g., CinePile (Rawal et al., 2024), EgoSchema (Mangalam et al., 2023), MoVQA (Zhang et al., 2023b), MLVU (Zhou et al., 2024), Video-MME (**?**)), to hour-scale evaluations (e.g., LVBench (Zhang et al., 2025), LONGVIDEOBENCH (Wu et al., 2024), HourVideo (Chandrasegaran et al., 2024), ALLVB (Tan et al., 2025), HLV-1K (Zou et al., 2025)). However, despite this progress in video duration, current long-video benchmarks are ill-equipped to properly assess the multi-timescale understanding of Multimodal Large Language Models (MLLMs). By probing different temporal scales using entirely different videos (Zhou et al., 2024) or relying on isolated video segments (Li et al., 2024d), these benchmarks inherently conflate a model's temporal reasoning with its response to varying content. This makes it exceedingly difficult to disentangle an MLLM's true performance at each specific timescale from content-driven artifacts, thereby obscuring a clear view of its temporal processing capabilities. ScaleLong addresses this critical limitation with its novel within-content, multi-scale design. By embedding questions at four hierarchical levels—from clip to story—within each individual video, it enables a precise, disentangled evaluation of an MLLM's capabilities across the temporal hierarchy. This approach not only reveals performance trends with longer contexts but also pinpoints specific strengths and failure modes at distinct granularities, offering a more rigorous methodology for assessing long-video understanding.

## 6    CONCLUSION

We introduce ScaleLong, the first benchmark to enable a fine-grained, within-content evaluation of MLLMs across hierarchical temporal scales. This design disentangles temporal reasoning from content variability, offering a more accurate assessment of multi-timescale understanding. Our evaluation of 23 MLLMs reveals a striking and consistent U-shaped performance curve: models excel at the shortest (Clip) and longest (Story) scales but falter at intermediate (Shot, Event) levels. This trend points to a systemic weakness in current modeling paradigms, which are adept at local feature extraction and global summarization but lack robust mechanisms for mid-range temporal coherence. Furthermore, our analysis indicates that strategically increasing visual token allocation offers a tangible path toward mitigating these deficits, though it is not a complete solution. ScaleLong's findings thus challenge the community to develop novel architectures and training objectives specifically designed to bridge this critical temporal gap. Ultimately, ScaleLong provides a crucial diagnostic

tool to benchmark progress and steer the development of MLLMs capable of genuine multi-scale long-video comprehension.

## 7 ETHICS STATEMENT

We have undertaken a comprehensive review of the ethical dimensions of this research, with a particular focus on data sourcing, potential biases, and labor practices. The ScaleLong dataset is composed entirely of videos sourced from YouTube, for which we employed a rigorous filtering process to select only content marked with a Creative Commons license, followed by a manual verification of each video's license status to ensure a legitimate copyright basis for academic research. Concurrently, we acknowledge the inherent limitations and potential biases of this approach; as our data is primarily sourced from a platform dominated by Western and English-speaking creators, ScaleLong may carry cultural, linguistic, and demographic skews. We transparently disclose this limitation to alert downstream users to the associated fairness risks and the possibility that models trained on this dataset may not generalize equitably across all populations. Furthermore, in all stages of this project involving human labor, including data annotation and evaluation, we adhered to fair labor practices. All contributors were compensated equitably in accordance with fair living wage standards for the region, to respectfully acknowledge their contributions.

## 8 REPRODUCIBILITY STATEMENT

We are fully committed to the reproducibility of our work. To facilitate verification of our findings, all code required to reproduce our experiments is public. Furthermore, the ScaleLong dataset is publicly and readily accessible via the Hugging Face Hub at: https://huggingface.co/datasets/ScaleLong/ScaleLong. In the Experiments section of our paper, we provide a detailed description of all experimental settings to ensure faithful replication such as the number of frames sampled per video, input resolution, and our evaluation protocols. We are confident that these comprehensive resources will allow for the straightforward reproduction of our experimental results.

## 9 ACKNOWLEDGEMENTS

Shiwen Ni was supported by GuangDong Basic and Applied Basic Research Foundation (2023A1515110718 and 2024A1515012003), and Shenzhen Science and Technology Program (JCYJ20250604182917023).

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

# A  APPENDIX

## A.1  ANNOTATION TUTORIAL

### A.1.1  QUESTION TYPE

---

**🔧 Question Type Details**

Representative examples of all 5 task categories in ScaleLong are shown in Figure 1.

**Causal Reasoning**

- These questions aim to test the model's ability to infer causal relationships between events, actions, or phenomena in the video. The model needs to understand the content of frames over a certain time segment, and identify the internal logic of the "cause-effect" chain.

**Object Recognition**

- These questions aim to assess the model's ability to identify specific objects, scenes, and their features (such as color, shape, and state) in the video. The model needs to locate them within the video scenes, achieve cross-frame tracking and consistent recognition.

**Action Understanding**

- These questions aim to test the model's ability to identify character actions or object movements in the video. The model needs to understand the temporal combination of actions and their semantic goals.

**Information Summary**

- These questions aim to test the model's ability to summarize or generalize the main content or details of the video. The model needs to use clues or context within the video to go beyond the understanding of individual frames or segments, grasp the core content of the video, and extract the plot summary.

**Counting Problems**

- These questions aim to assess the model's ability to conduct quantitative analysis of the number of objects, frequency of events, and temporal relationships in the video. The model needs to accurately identify and distinguish various elements, involving counting across multiple dimensions such as objects, plot elements, or actions.

---

### A.1.2  DATA ANNOTATION STEPS

---

**🔧 Operations:**

1. **Attributes that need to be annotated:**
   - **Video Key:** The video id in Youtube.
   - **Video Type:** Based on the content of the video, select one category from TV, sport, live, self-media, and documentary.
   - **Question Stems and Options:** Use clear and concise language to describe the question, ensuring that the question stem is explicit and specific. Each QA should include 4 options that are logically coherent and relevant to the question.
   - **Answer:** Provide a unique and correct answer.
   - **Question Type:** Select one of the 5 question types that align with the question stem, each question can have only one question type.
   - **Time Reference:** Label the time segment in the video corresponding to the answer (the time segment format should be in string format "XX:XX-XX:XX").
   - **Hierarchy:** Label the level to which the question belongs (clip, shot, event, or story).

2. **Watch Video and Determine Hierarchy Type:**
   - After fully viewing the video content to be annotated, select the two most appropriate and valuable question types for each hierarchy. Then, pre-conceive the corresponding question content in preparation for designing the question stems and distractors.

---

3. **Design Question Stem and Answer:**
   - **Question Design Requirements:**
     - **Clear Expression:** Ensure that the questions are concise and straightforward, avoiding complex or lengthy expression.
     - **Explicit Description:** Describe the core elements in the video clearly and specifically (such as scenes, characters, objects, actions, or weather), ensuring questions are unambiguous and refer to a unique segment in the video.
     - **Quantity Requirement:** For each video, design and annotate 2 questions for each hierarchy (clip, shot, event, story), need a total of 8 questions per video.
     - **Balanced Question Types:** Maintain a similar number of questions for each type (e.g., Causal Reasoning, Object Recognition, etc.).
     - **Target Visual Information:** Ensure that pure text-based LLMs cannot answer the questions correctly.
   - **Answer Design Requirements:**
     - **Uniqueness:** For each question, there must be a unique and clearly correct answer.
     - **Concise Language:** The wording of answer should be concise and clear, avoiding complex sentence structures and uncommon vocabulary.

4. **Design Distractor Options:** Design three incorrect distractors based on the question and correct answer. The distractors should also be described clearly, be of consistent length, and be meaningful.

5. **Option Format Requirements:**
   - **Consistent Length:** Ensure that four options' length are similar, avoid making the correct option easily identifiable due to length differences.
   - **Diverse Design:** Design distractors in a varied manner to avoid patterns, avoid consistently employing a singular approach when design.
   - **Concise Language:** The words of distractor options should be concise and clear, avoiding complex sentence structures and uncommon vocabulary.
   - **Option significance:** The distractors should be of the same category as the correct answer and should be meaningful in relation to the question stem.

### A.1.3 METHODS FOR DESIGNING INCORRECT ANSWER OPTIONS

🔧 **Incorrect Option Design Methods**

- **Visual Replacement:** Replace a piece of visual information in the video with information that is similar and incorrect. For example, altering the color or shape of an object.
- **Quantitative Replacement:** Change the quantity of a detail in the video.
- **Action Replacement:** Describe an action that is similar but different from the actually occurred action.
- **Character Replacement:** Associate the actual event with the wrong character.
- **Spatial Replacement:** Incorrectly describe the location where an event occurs.
- **Temporal Replacement:** Change the time point of an event in the description.
- **Missing Information:** Create an error by omitting key details in the option. (e.g. leaving out an important action or cause when describing an event)
- **Detail Replacement:** Manufacture an error by exaggerating or minimizing a detail. (e.g. describing "running slowly" as "sprinting quickly")
- **Sequential Replacement:** Arrange a series of events that occurred in the video in the wrong order.
- **Frequency Replacement:** Repeat the frequency of actions incorrectly.

### A.2 MANUAL REVIEW PROCESS DETAILS

The quality inspection of ScaleLong comprises two rounds.

- **Round 1:** focuses on standardizing problem structures and correcting elements that do not align with the question.

- **Round 2:** addresses advanced quality requirements to ensure task rigor.

### A.2.1 ROUND 1 QUALITY CONTROL

> **⊘ Purpose**
>
> - Ensure the comprehensiveness of the basic structure of the question and the alignment among each elements, including a clear question stem, a correct answer, and meaningful distractors.

> **❶ Quality Assessment Dimensions**
>
> 1. **Question Stem:**
>    - **Expression:** Check whether the question stem is coherent and meaningful.
>    - **Duplicate question stem:** Check whether the question content for the same video are repetitive, if only change the words of question stem consider as the same question.
>    - **Question Type:** Check whether the question type is correct and whether the question stem corresponds to the question type.
>    - **Temporal distribution:** Check whether the question stem only focuses on the specific time segments of the video.
> 2. **Options:**
>    - **Answer:** Check the correctness of the answer based on the original video.
>    - **Distractors Type:** Check whether the method of designing incorrect options same to the content of the incorrect options.
>    - **Distractors content:** Check whether the distractors are meaningful. Distractors should be as same category as the answer and should be meaningful in relation to the question stem.
> 3. **Absolute Time:** Replace question stems that use absolute time references with vague time references.
> 4. **Hierarchy:** Check whether the hierarchy is correctly labeled.

### A.2.2 ROUND 2 QUALITY CONTROL

> **⊘ Purpose**
>
> - Enhance task validity by verifying multimodal necessity, question stem accurity, and distractor plausibility, thereby preventing evaluation biases caused by design flaws.

> **❶ Methods**
>
> 1. **Information Leakage Detection:** Ensure that the question stem or options do not directly disclose the answer. (e.g. avoid using clothes color to locate a person when asking about the color of his clothes)
> 2. **Commonsense Dependency Screening:** Check whether the content of the question is common knowledge that can be answer directly and whether the question requires prior knowledge to answer. (e.g. "where does the sun rises", "What action did [Trump] take?").
> 3. **Duplicate question types:** Check whether the question types in a video are overly concentrated, and reannotate videos that are overly concentrated.
> 4. **Distractor Optimization:** Redesign meaningless distractors based on video content. Ideally, distractors should correspond to the question categories and create confusion.
> 5. **Video Quality Filtering:** Remove low-quality videos (difficult to describe with precise language) that cannot support effective question design.

A.2.3 RETENTION STATISTICS AND INTER-ANNOTATOR AGREEMENT

To quantify the rigor of our manual review process, we track data retention rates and measure inter-annotator consistency throughout the quality control pipeline.

**Verification Protocol:** We implement a strict **"1-Generation, 2-Verification"** protocol. Each question is initially generated by one **annotator** and subsequently verified by two independent **annotators**. We enforce a **unanimous consensus rule**: a sample is retained only if both independent verifiers agree on its validity, including the accuracy of timescale categorization and answer correctness.

**Inter-Annotator Agreement:** We utilize Fleiss' Kappa ($\kappa$) to evaluate the consistency between annotators throughout the quality control pipeline. The process demonstrates robust **annotator** consensus across both stages: **Round 1 achieves** $\kappa = 0.85$ for basic structural correctness and temporal scale alignment, while **Round 2 reaches an almost perfect** $\kappa = 0.90$ for advanced quality criteria such as information leakage detection and distractor plausibility. These consistently high metrics confirm that our rigorous filtering criteria are applied uniformly, ensuring the reliability of the final benchmark.

**Data Retention Analysis:** The rigorous two-round filtering process results in a highly curated dataset. As shown in Table 3, the overall question retention rate is 54.6%, reflecting our high standard for quality assurance.

Table 3: Statistics of data retention and inter-annotator agreement. Fleiss' Kappa is calculated independently for each verification round based on the agreement between the two verifiers.

| QC Stage | Focus | Retained Volume | | Step Retention | | Kappa ($\kappa$) |
|----------|-------|--------|-----------|--------|-----------|-----------|
| | | Videos | Questions | Videos | Questions | |
| **Round 1** | Clarity & Timescale | 331 | 2,430 | 82.8% | 75.9% | **0.85** |
| **Round 2** | Leakage & Distractors | 269 | 1,747 | 81.3% | 71.9% | **0.90** |
| **Final** | **Total / Cumulative** | **269** | **1,747** | **67.3%** | **54.6%** | – |

A.3 TIMESCALE ALIGNMENT VERIFICATION

To ensure the robust alignment of our questions with their intended timescales, we implement a rigorous annotation protocol, empirically validated through comprehensive experiments. These experiments unequivocally demonstrate both the **sufficiency** of our designated timescales—showing no additional performance benefit from expanded temporal contexts—and their **necessity**—evidencing significant performance degradation when temporal contexts are restricted.

A.3.1 ANNOTATION PROTOCOL FOR TIMESCALE DEFINITION

Our annotation protocol is founded upon clearly defined timescales, meticulously applied by highly qualified annotators (Ph.D.s and Ph.D. candidates). To maintain the highest quality and consistency, each question undergoes a thorough review process and is only accepted after successfully passing a stringent peer-review check, requiring independent approval from two additional annotators. This multi-stage validation ensures the precision and reliability of our timescale annotations.

A.3.2 SUFFICIENCY OF TIMESCALE (UP-SCALING EXPERIMENT)

To empirically validate the sufficiency of our annotated timescales, we conduct an up-scaling experiment using the InternVL-2.5-8B model. This experiment involves evaluating questions by expanding their associated video segments to the next higher timescale (e.g., questions originally annotated for a 'Clip' scale are tested on 'Shot' segments). We quantify the change in performance using the accuracy difference metric, $\Delta Acc_{up}$:

$$\Delta Acc_{up} = Acc_{expanded} - Acc_{original} \tag{1}$$

Where $Acc_{original}$ denotes the accuracy obtained when the model is tested using the video segment corresponding to the originally annotated time span, and $Acc_{expanded}$ represents the accuracy when the model is provided with the expanded temporal context.

The results, presented in Table 4, indicate that expanding the temporal context yields only marginal performance improvements. The small positive $\Delta Acc_{up}$ values confirm that our original timescale annotations already encapsulate the essential visual information required to answer the questions effectively, thereby establishing the sufficiency of our defined timescales.

Table 4: Up-scaling Experiment Results

| Up-Scaling | $\Delta Acc_{up}$ |
|---|---|
| Clip $\rightarrow$ Shot | +1.6% |
| Shot $\rightarrow$ Event | +1.2% |
| Event $\rightarrow$ Story | +2.0% |

### A.3.3 NECESSITY OF TIMESCALE (DOWN-SCALING EXPERIMENT)

To establish the necessity of our annotated timescales—demonstrating that long-timescale questions cannot be adequately answered with only local information—we perform a down-scaling experiment with the InternVL-2.5-8B model. In this setup, questions are evaluated on significantly shorter video segments extracted using a sliding window approach. To ensure a fair comparison, experiments are conducted across multiple shorter segments, and the best achievable performance is selected.

We employ two key metrics to quantify the observed performance degradation:

1. **Absolute Performance Difference ($\Delta Acc_{down}$):** This metric quantifies the absolute performance loss when the original timescale video segment is downscaled.

$$\Delta Acc_{down} = Acc_{original} - Acc_{best\_downscaled} \tag{2}$$

   Where $Acc_{original}$ is the model's accuracy on the original annotated timescale segment, and $Acc_{best\_downscaled}$ is the best accuracy achieved using the shorter, downscaled video segments.

2. **Visual Gain Retention Ratio (VGRR):** This metric measures the proportion of performance gain, initially derived from the original full-timescale video segment, that is retained after down-scaling.

$$VGRR = \frac{Acc_{best\_downscaled} - Acc_{text\_only}}{Acc_{original} - Acc_{text\_only}} \times 100\% \tag{3}$$

   Where $Acc_{text\_only}$ is the baseline accuracy when the model answers with no video input, relying solely on textual priors.

The results, summarized in Table 5, clearly demonstrate a substantial performance degradation. The large positive values for $\Delta Acc_{down}$ indicate a significant loss in absolute accuracy, while the low VGRR values confirm that the majority of critical visual information and context provided by the original, broader timescale is lost during down-scaling. This provides strong evidence that a broader temporal context is indeed necessary for answering long-timescale questions effectively.

Table 5: Down-scaling Experiment Results

| Down-scaling | $\Delta Acc_{down}$ | VGRR |
|---|---|---|
| Shot $\rightarrow$ Clip | 15.1% | 31.1% |
| Event $\rightarrow$ Shot | 15.0% | 25.0% |
| Story $\rightarrow$ Event | 11.0% | 33.1% |

### A.3.4 CONCLUSION ON TIMESCALE ALIGNMENT

In conclusion, our stringent annotation protocol, rigorously validated by quantitative up-scaling and down-scaling experiments, unequivocally confirms that our questions are correctly aligned with their intended timescales. The up-scaling experiments establish the sufficiency of our annotations (demonstrating no significant benefit from adding more context), while the down-scaling experiments confirm their necessity (highlighting that removing context significantly harms performance). This robust methodology guarantees that questions designed to necessitate a broad understanding of video content cannot be trivially solved with only local temporal information, thus validating the integrity of our dataset's inherent timescale hierarchy.

## A.4 SENSITIVITY ANALYSIS OF DISTRACTOR LENGTH BIAS

We acknowledge the importance of mitigating annotation artifacts, specifically the potential for models to exploit answer length as a heuristic. To rigorously validate the integrity of our benchmark, we provide comprehensive statistical and experimental evidence demonstrating that our distractors are well-balanced and that model performance is not driven by length cues.

### A.4.1 STATISTICAL ANALYSIS OF LENGTH DISTRIBUTION

We conducted a systematic analysis of answer and distractor lengths across all 1,747 questions using character count as the metric. The results indicate a negligible length bias: the **Answer-to-Distractor Length Ratio is 1.083**. This implies that correct answers are, on average, only 8.3% longer than incorrect options, suggesting that length is not a salient statistical signal for distinguishing the correct answer.

### A.4.2 EXPERIMENTAL VALIDATION VIA LENGTH-CONTROLLED SUBSETS

To further examine whether models systematically exploit length cues, we constructed two diagnostic subsets with contrasting distributions:

- **Length-Balanced Subset ($n = 1,148$):** Constructed via stratified sampling to ensure equal representation across all length ranks (25% per quartile), thereby eliminating systematic length bias.

- **Length-Imbalanced Subset ($n = 599$):** Adversarially constructed such that 94.3% of the correct answers are the longest option, maximizing the exploitability of length-based heuristics.

We evaluated six state-of-the-art MLLMs on both subsets. The comparative results are reported in Table 6.

Table 6: Performance comparison on Length-Balanced and Length-Imbalanced subsets. The small performance gap ($\Delta$) indicates minimal reliance on length heuristics.

| Model | Balanced Subset | Imbalanced Subset | $\Delta$ Performance |
|---|---|---|---|
| Gemini-2.5-Pro | 68.1% | 67.6% | +0.5% |
| Doubao-1.5 | 56.0% | 61.8% | -5.8% |
| InternVL2.5-78B | 54.8% | 61.1% | -6.3% |
| GPT-4o | 53.2% | 58.0% | -4.8% |
| LLaVA-Video | 49.6% | 54.4% | -4.8% |
| Aria | 47.8% | 53.6% | -5.9% |
| **Mean** | **54.9%** | **59.4%** | **-4.5%** |

### A.4.3 PRINCIPAL FINDINGS

Our analysis yields three key observations that confirm the validity of the benchmark:

1. **Negligible Performance Differential:** The mean performance gap between the balanced and maximally exploitable (imbalanced) subsets is merely 4.5%. This indicates that length cues confer minimal advantage, and models are primarily relying on visual-semantic understanding.

2. **Robustness of Leading Models:** Notably, the top-performing model, *Gemini-2.5-Pro*, achieves superior performance on the Length-Balanced subset (68.1%) compared to the Imbalanced subset (67.6%). This inverse behavior definitively excludes systematic length-based exploitation as a primary strategy for advanced models.

In conclusion, both statistical metrics and experimental stress-testing confirm that our benchmark design effectively suppresses distractor length bias, ensuring that the evaluation reflects genuine multimodal reasoning capabilities.

### A.5 MORE EXAMPLES

To provide a clearer understanding of the key timescales proposed in this paper, we present a detailed specification of our annotation standards. We categorize temporal granularity into four distinct levels—Clip, Shot, Event, and Story—based on their duration range and the visual scope required for reasoning. Table 1 outlines these specific criteria and decision rules. Furthermore, we include two illustrative examples to further elucidate the distinctions between these four different timescales.

Table 7: ScaleLong Annotation Criteria & Decision Rules

| Timescale | Duration Range | Decision Rule / Visual Scope |
|---|---|---|
| **Clip** | < 3 Seconds | **Single Moment:** Solvable by analyzing a few consecutive frames. |
| **Shot** | 4 – 15 Seconds | **Single Continuous Shot:** Requires analyzing information within one uninterrupted camera take. |
| **Event** | 16 Sec – 10 Min | **Multi-Shot Sequence:** Spans multiple consecutive shots but remains within a localized context. |
| **Story** | > 10 Minutes | **Global Narrative:** Covers the entire video or substantial portions; cannot be solved by a single sequence. |

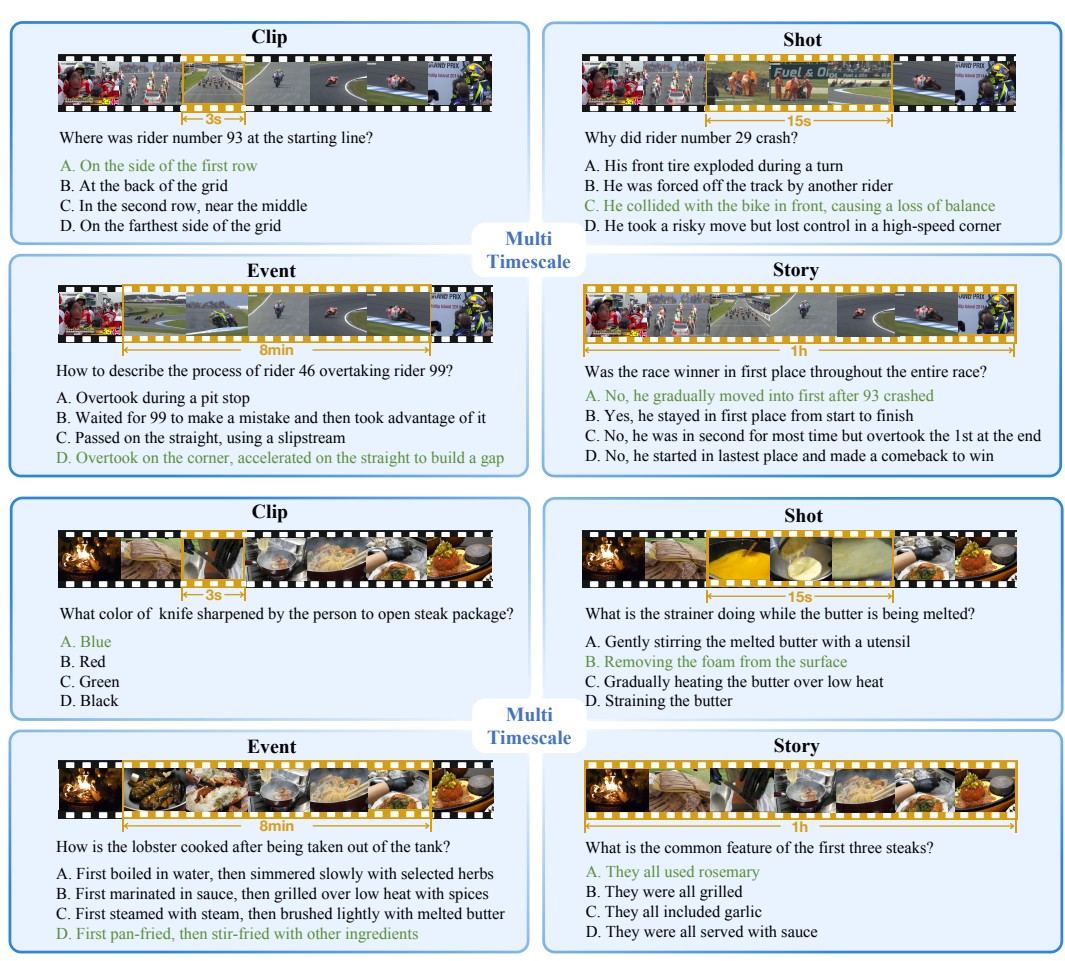

Figure 5: More representative samples from ScaleLong. The top and bottom panels illustrate different examples, where each sample comprises a video paired with carefully designed questions, structured across four hierarchical temporal scales. The correct answers are indicated in green.

