# OpenReview forum: "ScaleLong: A Multi-Timescale Benchmark for Long Video Understanding"
_ICLR.cc/2026/Conference — ICLR 2026 Poster_

### Official Review · Reviewer_dYA8 · 2025-10-25

**Soundness:** 3
**Presentation:** 3
**Contribution:** 3
**Rating:** 6
**Confidence:** 4

**Summary:**

This paper proposes ScaleLong, a new benchmark designed to evaluate multi-modal large language models on long video understanding across multiple temporal scales. Unlike prior benchmarks that either use short clips or do not explicitly disentangle temporal granularity, ScaleLong embeds four levels of questions within the same video, including clip, shot, event, and story. It enables direct comparison across timescales. The benchmark includes 269 long videos (avg. length 86 minutes) from 5 main- and 36 sub-categories, each with 4-8 carefully designed questions. Extensive experiments on 23 commercial and open-source MLLMs reveal a consistent U-shaped performance trend, where models perform well on the shortest and longest timescales but struggle at intermediate ones. The authors also provide ablation studies indicating that increasing visual token capacity improves performance.

**Strengths:**

1. Overall, the problem addressed is well-motivated: existing benchmarks cannot reliably disentangle model performance across different temporal scales in long videos, and the within-content question design is a meaningful contribution.
2. The dataset construction methodology is described clearly, including multi-stage annotation and quality control to ensure question grounding and discourage guessable answers.
3. The experimental study is thorough, covering a diverse set of 23 models spanning multiple sizes and both closed & open-source settings, which makes the analysis convincing.
4. The discovery of a U-shaped performance curve across timescales is interesting and insightful, and is further supported by human baselines for difficulty validation.

**Weaknesses:**

1. My major concern is the size of the benchmark. It contains 269 videos, which is slightly smaller than other recent long-video benchmarks, and the paper mostly argues for "quality over scale". It might be useful to include more justifications, e.g., a statistical analysis of annotation diversity, to demonstrate the effectiveness and significance of such a small benchmark.
2. The U-shaped trend is an important and interesting observation, but the explanation provided is somewhat speculative (e.g., "models excel at short and global context but struggle mid-term"). More controlled experiments to identify why mid-level reasoning fails would strengthen the conclusion.
3. Given the observed U-shaped trend, I believe it might be also interesting to investigate more types of video understanding models, e.g., grounded video understanding methods [1-3] and agentic solutions [4-5], rather than running experiments solely with end2end methods. More discussions on the performance trend of different types of models shall provide more insights to the readers.

[1] SeViLA: Self-Chained Image-Language Model for Video Localization and Question Answering
[2] VideoChat-TPO: Task Preference Optimization: Improving Multimodal Large Language Models with Vision Task Alignment
[3] VideoMind: A Chain-of-LoRA Agent for Long Video Reasoning
[4] VideoAgent: Long-form Video Understanding with Large Language Model as Agent
[5] VideoAgent: A Memory-augmented Multimodal Agent for Video Understanding

**Questions:**

Please refer to the weakness section for my questions. Overall, I leaning to accepted this paper given the considerable quality.

---

> ### Author Response · Authors · 2025-11-26
>
> > ### W1: Benchmark Size and Justification for "Quality over Scale"
>
> We thank the reviewer for raising this important concern regarding the benchmark size. While our dataset contains 269 videos, which is numerically smaller than some short-video benchmarks, we argue that for the specific domain of **long-video understanding**, this scale is not only sufficient but represents a significant contribution when considering total duration, annotation density, and evaluation practicality.
>
> **1. Inherent Trade-off in Extreme Long Video Benchmarks:**
>
> As highlighted in recent literature, notably LVBench [1], constructing high-quality long-video benchmarks involves an unavoidable trade-off between video duration/quality and total video count. The scarcity of long video understanding benchmarks can be attributed to the challenges in data collection and the complexity of annotation. Unlike short videos that can be mass-processed, long videos require annotators to watch and analyze hours of content to design valid questions, creating a massive annotation bottleneck.
>
> Despite these challenges, ScaleLong significantly advances the scale boundary in this domain. Compared to LVBench, which contains **103 videos** (average 68 min), ScaleLong provides **269 videos** with an even longer average duration of **86 minutes**—representing a dataset that is **2.6x larger** in video count and significantly larger in total duration. Moreover, our annotation complexity is substantially higher: unlike other benchmarks, ScaleLong requires every single video to be annotated with questions covering **all four temporal scales** (Clip, Shot, Event, Story). This dense, hierarchical annotation requirement means we must select videos rich enough to support questions at all granularities—from 3-second details to 86-minute narrative arcs—making each video sample in ScaleLong exceptionally information-rich and diagnostically valuable.
>
> **2. Evaluation Efficiency and Practicality:**
>
> For long-video benchmarks, evaluation cost is a critical practical constraint. Processing an 86-minute video consumes vastly more computational resources (tokens, inference time) than processing a 1-minute clip. A benchmark with thousands of hour-long videos would be prohibitively expensive for the research community to use routinely. With ~270 videos and ~1,700 questions, ScaleLong offers a balanced "sweet spot": it provides sufficient statistical power (as demonstrated in our W2 response on sample size reliability) while remaining computationally feasible for widespread adoption and iterative model development.
>
> [1] Wang W, He Z, Hong W, et al. Lvbench: An extreme long video understanding benchmark[C]//Proceedings of the IEEE/CVF International Conference on Computer Vision. 2025: 22958-22967.

---

> ### Author Response · Authors · 2025-11-26
>
> > ### W2: Causal Analysis of the U-shaped Performance Trend
>
> We thank the reviewer for this insightful comment. We agree that attributing the "U-shaped" trend (where models perform better at Clip/Story extremes but struggle at Shot/Event mid-ranges) requires more than speculative reasoning. To provide concrete empirical evidence, we conducted a **fine-grained error analysis** across five representative models (GPT-4o, Gemini-1.5-Pro, InternVL2.5, Doubao-1.5-Pro, LLaVA-Video).
>
> **Average Error Rate  by Task Type and Timescale:**
>
> The table below presents the average error rate for each specific error type across the four temporal scales. This granular breakdown isolates the specific failure modes driving the performance drop at intermediate scales.
>
> | Error Type | Video Clip | Video Shot | Video Event | Video Story |
> | :--- | :--- | :--- | :--- | :--- |
> | **Visual Replacement** | 23.3% | 33.9% | **44.6%** | 37.1% |
> | **Spatial Replacement** | 30.4% | 33.0% | **45.7%** | 10.9% |
> | **Missing Information** | 7.5% | 19.1% | **33.5%** | 13.6% |
> | **Action Replacement** | 33.7% | **41.2%** | 28.5% | 23.4% |
> | **Character Replacement** | 39.4% | **45.4%** | 26.1% | 27.3% |
> | **Frequency Replacement** | N/A | **71.8%** | 62.9% | 57.3% |
> | **Temporal Replacement** | 24.6% | 37.7% | 41.0% | **48.9%** |
> | **Quantitative Replacement** | 51.5% | 54.1% | 56.0% | **57.5%** |
>
> **Empirical Findings:**
>
> This data reveals distinct "bottleneck mechanisms" that explain the U-shaped curve:
>
> 1.  **Why Event-level fails:**
>     *   **Visual & Spatial Collapse:** The error rates for *Visual Replacement* (44.6%) and *Spatial Replacement* (45.7%) peak specifically at the Event level. This indicates that models struggle most to maintain consistent tracking of visual objects and their spatial relationships when the context expands from a single shot to multi-shot sequences.
>     *   **Information Overload:** The peak in *Missing Information* errors (33.5%) at the Event level suggests that models fail to filter critical details when integrating information across multiple shots, leading to "lost in the middle" hallucinations.
>
> 2.  **Why Shot-level fails:**
>     *   **Action & Character Confusion:** *Action Replacement* (41.2%) and *Character Replacement* (45.4%) are highest at the Shot level. This confirms that intermediate-duration continuous sequences pose the greatest challenge for fine-grained dynamic recognition, likely due to the loss of temporal resolution in sparse sampling which affects motion continuity more than static object recognition.
>
> 3.  **Why Clip/Story perform better.**
>     *   **Clip (Local Peak):** Benefits from low error rates in static tasks, as recognition requires minimal temporal integration.
>     *   **Story (Global Peak):** Interestingly, perceptual error rates (Visual, Action, Character, Spatial) **drop significantly** at the Story level. This suggests that the "Global Context" acts as a regularizer—models can leverage long-range narrative patterns and redundancy to correct local perceptual errors, explaining the performance recovery at the longest timescale.

---

> ### Author Response · Authors · 2025-11-26
>
> > ### W3: Expanding Evaluation to Grounded and Agentic Models
>
> We thank the reviewer for this constructive suggestion. We fully agree that evaluating a broader spectrum of methodologies, particularly **grounded video understanding models [1-3]** and **agentic solutions [4-5]**, would provide valuable comparative insights into the "U-shaped" performance trend. Contrasting these modular or iterative approaches with end-to-end models could indeed reveal whether explicit grounding or agentic planning can mitigate the specific reasoning bottlenecks we identified at the Shot/Event levels.
>
> Due to the limited time and computational resources available during the rebuttal phase, we were unable to complete the full adaptation and evaluation of these distinct model architectures on ScaleLong. However, we recognize the significant value of this comparison. We are currently working on setting up the evaluation pipelines for these methods and will do our utmost to include their performance results and a comparative discussion in the final version of the manuscript to provide a more comprehensive landscape of long-video understanding capabilities.

---

### Official Review · Reviewer_41fk · 2025-10-26

**Soundness:** 2
**Presentation:** 3
**Contribution:** 2
**Rating:** 4
**Confidence:** 4

**Summary:**

This paper proposes ScaleLong, a benchmark for measuring the model's ability to understand content at different temporal scales (Clip, Shot, Event, and Story) in long videos. It uses five real-world video categories collected from YouTube, with an average video length of over one hour. This paper comprehensively tests existing MLLMs (both open-source and closed-source) on ScaleLong, revealing some of their shortcomings.

**Strengths:**

1. The motivation for this study is sound, and the comprehensive measurement of the model's ability to perceive and understand temporal information at different scales in long videos is meaningful.
2. The paper is clearly organized and easy to understand.

**Weaknesses:**

1. Long videos collected from YouTube, particularly TV and sports videos, often feature famous events. The training corpus for MLLMs may contain information about these events. How can the authors ensure that the questions they ask are not answered by the inherent knowledge contained in the model?
2. Where do the four time scales of Clip (less than 3 seconds), Shot (4-15 seconds), Event (16-10 minutes), and Story (more than 10 minutes) and their respective time range division standards come from?
3. (1) If the upper limit of the model input is 256 frames, for an 89-minute video, the average time range contained in each frame will reach 89*60 / 256 = 20.85 seconds, which means that many CLIPs and Shots will be ignored when sampling video frames, while MLLMs performs relatively well at the Clip-level time scale. Does this mean that many Clip-level questions are related to a large number of clips in the video? Therefore, can they be truly called Clip-level questions? (2) On the contrary, some Shot and Event-level questions may be related to dense spatio-temporal motions, so sparse frame sampling cannot cover the complete spatiotemporal clues, resulting in low model performance.
4. In addition, can part of the reason why counting questions cannot be answered well be attributed to insufficient input frames? After all, sparse video content sampling may cause the time when objects appear to be ignored? The authors should further explore the impact of the number of input frames on different question types and timescales.

**Questions:**

1. How is the error defined and measured in Section 4.4, such as missing information and spatial replacement in L425?
2. New models, such as InternVL3 and InternVL3.5, could also be tested before the paper is submitted.

In summary, I believe the benchmark's design is sound, but some details concern me. I would be happy to discuss the points raised in the weaknesses and questions with the authors.

---

> ### Author Response · Authors · 2025-11-26
>
> > ### W1: Data Contamination from Famous Events
>
> We appreciate this important concern about potential data contamination. While our videos are collected from public platforms like YouTube and may contain well-known events, we have taken several measures to ensure the validity of our evaluation:
>
> **1. Original question design.**
>
> Although the video content itself may have appeared in model training data, all questions and answer options in ScaleLong are specifically designed by us and are unlikely to exist in any training corpus. Our questions require models to watch the video and understand specific temporal segments to answer correctly, rather than relying on memorized knowledge about the events.
>
> **2. Standard practice in the field.**
>
> We follow the standard practice adopted by existing long-video benchmarks, including MLVU [1], LVBench [2], VideoMME [3], which all use publicly available videos. This approach is widely adopted because public videos provide rich content diversity and realistic scenarios.
>
> **3. Text-only baseline validation.**
>
> To empirically verify the extent to which models rely on memorized knowledge versus video content, we conduct text-only experiments where we provide only the question text to the model without any video input. Testing InternVL3.5-8B under this condition yields an accuracy of 37.7%, compared to the random baseline of 25% for 4-choice multiple-choice questions. This 12.7 percentage point elevation above random indicates that models can leverage general world knowledge to eliminate some implausible distractors; however, this improvement is relatively modest and controlled. The limited performance gain suggests that while models may possess some prior knowledge about common events or scenarios, the benchmark does not suffer from severe data contamination that would enable models to answer questions through pure memorization. We acknowledge that minor knowledge leakage may exist for certain well-known events; however, the controlled magnitude of text-only performance (only 12.7 points above random) demonstrates that this effect is relatively limited and does not fundamentally compromise the benchmark's validity in assessing genuine video understanding capabilities.
>
> These factors collectively ensure that ScaleLong evaluates genuine video understanding capabilities rather than memorized knowledge.
>
> [1] Zhou J, Shu Y, Zhao B, et al. Mlvu: Benchmarking multi-task long video understanding[C]//Proceedings of the Computer Vision and Pattern Recognition Conference. 2025: 13691-13701.
>
> [2] Wang W, He Z, Hong W, et al. Lvbench: An extreme long video understanding benchmark[C]//Proceedings of the IEEE/CVF International Conference on Computer Vision. 2025: 22958-22967.
>
> [3] Fu C, Dai Y, Luo Y, et al. Video-mme: The first-ever comprehensive evaluation benchmark of multi-modal llms in video analysis[C]//Proceedings of the Computer Vision and Pattern Recognition Conference. 2025: 24108-24118.

---

> ### Author Response · Authors · 2025-11-26
>
> > ### W2: Origin of Temporal Scale Definitions
>
> We thank the reviewer for this important question. The four temporal scales in ScaleLong—Clip (≤3s), Shot (4-15s), Event (16s-10min), and Story (>10min)—are not arbitrarily determined but systematically derived through a principled approach that integrates theoretical grounding from existing video understanding literature with empirical analysis of our dataset's inherent temporal characteristics.
>
> **Theoretical Foundation in Video Understanding Literature:**
>
> The hierarchical organization of temporal information in video content represents a well-established principle in long-video understanding research. Contemporary benchmarks consistently adopt multi-level temporal structures with semantically comparable granularities, including NewsNet's hierarchical temporal segmentation [1], VidChapters and Chapter-LLaMA's chapter-level organization [2,3], and MLVU and LVBench's multi-granularity temporal divisions [4,5]. While the precise numerical boundaries vary across benchmarks, the field demonstrates consensus regarding the fundamental hierarchical progression from local frame-level cues to multi-shot event sequences and ultimately to global narrative structures. This convergent recognition validates the conceptual necessity of stratified temporal categorization in comprehensive video understanding evaluation.
>
> **Empirical Derivation from Data Distribution:**
>
> Building upon this theoretical foundation, we conduct systematic analysis of the temporal duration distributions inherent in our dataset. Specifically, we examine the actual temporal span required for answering each question, revealing natural clustering patterns with distinct breakpoints at approximately 3 seconds, 15 seconds, and 10 minutes. These empirically observed thresholds align precisely with semantically meaningful boundaries corresponding to local frame cues (Clip), single-shot continuous sequences (Shot), multi-shot event progressions (Event), and global narrative contexts (Story). The convergence between empirical data characteristics and established semantic boundaries in video understanding research validates the ecological validity of our temporal scale definitions.
>
> **Generalizability and Adaptability:**
>
> While we acknowledge that any discrete temporal categorization necessarily involves design decisions—a characteristic shared across all existing long-video benchmarks—our definitions are both empirically grounded in observed data patterns and theoretically aligned with established hierarchical structures in the literature. Critically, the fundamental insights generated by ScaleLong regarding multi-scale temporal understanding capabilities transcend the specific numerical boundaries employed. The observed performance patterns reflect intrinsic properties of how models process information across different temporal granularities, ensuring that our core findings remain valid and transferable even under alternative boundary specifications. Furthermore, our dataset maintains flexibility for adaptation to different temporal scale taxonomies: researchers can re-annotate or re-categorize questions according to alternative temporal hierarchies tailored to specific research objectives, as the underlying video content and question design remain independent of any particular scale definition. This adaptability ensures that ScaleLong serves as a reusable resource for investigating multi-scale temporal understanding under diverse analytical frameworks.
>
> **References:**
>
> [1] Wu H, Chen K, Liu H, et al. Newsnet: A novel dataset for hierarchical temporal segmentation[C]//Proceedings of the IEEE/CVF Conference on Computer Vision and Pattern Recognition. 2023: 10669-10680.
>
> [2] Yang A, Nagrani A, Laptev I, et al. Vidchapters-7m: Video chapters at scale[J]. Advances in Neural Information Processing Systems, 2023, 36: 49428-49444.
>
> [3] Ventura L, Yang A, Schmid C, et al. Chapter-Llama: Efficient Chaptering in Hour-Long Videos with LLMs[C]//Proceedings of the Computer Vision and Pattern Recognition Conference. 2025: 18947-18958.
>
> [4] Zhou J, Shu Y, Zhao B, et al. Mlvu: Benchmarking multi-task long video understanding[C]//Proceedings of the Computer Vision and Pattern Recognition Conference. 2025: 13691-13701.
>
> [5] Wang W, He Z, Hong W, et al. Lvbench: An extreme long video understanding benchmark[C]//Proceedings of the IEEE/CVF International Conference on Computer Vision. 2025: 22958-22967.

---

> ### Author Response · Authors · 2025-11-26
>
> > ### W3: Impact of Sparse Sampling on Scale-Specific Performance
>
> We thank the reviewer for this sharp quantitative analysis. We address the two parts of your concern regarding Clip-level validity and Shot/Event-level performance drops below.
>
> **(1) Validity of Clip-level Questions under Sparse Sampling:**
>
> You are correct that with uniform sampling (e.g., ~20s interval), the probability of directly "hitting" a specific <3s clip is mathematically low. However, our empirical results, particularly the high performance of both humans and models, reveal that solving these questions does not strictly require capturing the exact frames of the instantaneous action.
>
> We attribute this phenomenon to three key factors:
>
> 1.  **Nature of Clip-level Questions (Visual Persistence):** Our analysis of task types reveals that **61.3%** (272/444) of Clip-level questions involve **Objective Recognition**. While a specific interaction (e.g., "holding a red cup") may define the clip boundary, the object itself (the red cup) often persists in the scene for a much longer duration than the 3-second action window. The sampler is highly likely to capture the object in adjacent frames, allowing the model to answer correctly based on visual persistence.
>
> 2.  **Information Diffusion vs. Annotation Validity:** One might question whether success under sparse sampling implies incorrect temporal annotations (i.e., the evidence lies outside the marked clip). We argue that this reflects the model's ability to utilize **contextual inference** rather than annotation error.
>     *   **Differentiation:** Our timestamps strictly mark the *exact occurrence* of the action (e.g., the moment of falling). However, sparse sampling often captures the **resultant states** (e.g., lying on the floor) or **preconditions** visible in adjacent frames.
>     *   **Consistency with Validation:** Our **Up-scaling experiments** (Appendix A.3) show that explicitly adding these surrounding frames (Clip → Shot) yields minimal performance gain (+1.6%) when the Clip is already available. This confirms that the **Clip itself contains the most direct and sufficient evidence** (validating our annotation). The fact that models can *also* succeed with only sparse surrounding frames demonstrates their robustness in inferring the missing primary event from secondary contextual cues, not that the primary event annotation was misplaced.
>
> 3.  **Empirical Validation via Human Performance:** The most compelling evidence comes from our **Controlled Human Frame-Sampling Experiment**. To directly assess whether sparse sampling makes these questions unanswerable, we conducted a controlled study where human annotators were restricted to the exact same sparse input (256 frames, uniform sampling) as the models. The results are presented below:
>
>     | Temporal Scale | Sparse Frames (Human) | SOTA Model (Gemini 2.5 Pro) |
>     |----------------|-----------------------|-----------------------------|
>     | **Clip**       | **77.1%**             | 71.5%                       |
>     | **Shot**       | **80.2%**             | 62.8%                       |
>     | **Event**      | **82.6%**             | 68.0%                       |
>     | **Story**      | **84.7%**             | 69.0%                       |
>
>     Humans achieved **77.1% accuracy** on Clip-level questions even under sparse sampling. This empirically proves that **sufficient information to answer these questions is preserved even under sparse sampling**, verifying that the questions are solvable through object persistence and contextual inference rather than requiring lucky frame hits.
>
> **(2) Disentangling Sampling Loss from Reasoning Deficiency in Shot/Event Levels:**
>
> We fully agree with the reviewer that sparse sampling inherently loses dense spatio-temporal clues, which is particularly detrimental for Shot and Event-level questions involving complex motion dynamics. However, our data suggests this is not the *sole* cause of low model performance:
>
> -   **The "Human-Model Gap" Evidence:** As shown in the table above, while sparse sampling limits the upper bound of information access, our controlled experiment reveals a massive **17.4% performance gap** at the Shot-level between humans (80.2%) and the best model (62.8%) under **identical sparse input conditions**.
> -   **Conclusion:** This indicates that while "missing frames" (input disparity) is a factor, the dominant failure mode is the model's inability to reason about the frames it *does* see (reasoning deficiency). Even when provided with the same sparse visual cues that allow humans to achieve >80% accuracy, models fail to reconstruct the inter-shot relationships and event progressions, highlighting a fundamental capability bottleneck in intermediate-scale temporal reasoning that goes beyond sampling density.

---

> ### Author Response · Authors · 2025-11-26
>
> > ### W4: Impact of Frame Sparsity on Counting Tasks and Frame Count Analysis
>
> We acknowledge that sparse sampling inherently limits fine-grained object tracking, which can indeed impact performance on counting tasks.
>
> **1. Analysis of Frame Count Impact:**
>
> As noted by the reviewer, we have already conducted comprehensive experiments analyzing model performance across different frame counts (32, 64, 128, 256) for each timescale in **Lines 353-368 of the main paper**. To further address your specific suggestion, we provide a detailed breakdown of performance by task type across varying frame counts (using InternVL2-8B as the base model).
>
> **Performance by Task Type vs. Frame Count:**
>
> | Task Type | 4 Frames | 8 Frames | 16 Frames | 32 Frames | 64 Frames | 128 Frames |
> | :--- | :--- | :--- | :--- | :--- | :--- | :--- |
> | Action Understanding | 42.10% | 43.90% | 45.30% | 45.50% | 46.00% | 48.40% |
> | Causal Reasoning | 43.50% | 43.30% | 49.40% | 47.30% | 52.20% | 51.00% |
> | Counting Problem | 31.70% | 34.00% | 35.90% | 35.00% | 34.60% | 39.00% |
> | Information Summary | 44.50% | 45.40% | 48.60% | 50.00% | 50.70% | 53.50% |
> | Object Recognition | 45.20% | 48.70% | 51.90% | 55.10% | 55.30% | 57.00% |
>
> **Observations:**
> The results demonstrate that increasing frame count consistently improves performance across most task types. Specifically for **Counting Problems**, performance improves from 31.70% (4 frames) to 39.00% (128 frames), confirming the reviewer's hypothesis that denser sampling aids counting. However, counting remains the most challenging task type even at higher frame counts, reflecting the intrinsic difficulty of tracking multiple entities over long durations.
>
> **2. Justification for Current Sampling Strategy:**
>
> Regarding the concern about counting tasks specifically, we offer three clarifications:
>
> - **Standard Community Practice:** Our uniform frame sampling strategy aligns with the established standard adopted by major long-video benchmarks (VideoMME [1], MLVU [2], LVBench [3], LongVideoBench [4]) and mirrors the training protocols of mllms (InternVL2.5 [5], Qwen2.5-VL [6], LLaVA-OneVision [7]). This methodological consistency ensures that our evaluation accurately reflects how models are designed to process visual information in both training and inference contexts.
>
> - **Maximizing Frame Utilization:**  we provision the majority of models with the maximum number of frames they can effectively process. We actively mitigate sampling limitations by pushing models to their limits.
>
> - **Intrinsic Model Capability:** The capacity to process dense frames is fundamentally constrained by a model's context window and computational efficiency. We contend that a model's inability to handle the frame density required for accurate counting is a reflection of its intrinsic architectural limitations, not merely an artifact of the evaluation protocol. Benchmarking models under these realistic constraints provides a more ecologically valid assessment of their current capabilities.
>
> **References:**
>
> [1] Fu C, Dai Y, Luo Y, et al. Video-mme: The first-ever comprehensive evaluation benchmark of multi-modal llms in video analysis[C]//Proceedings of the Computer Vision and Pattern Recognition Conference. 2025: 24108-24118.
>
> [2] Zhou J, Shu Y, Zhao B, et al. Mlvu: Benchmarking multi-task long video understanding[C]//Proceedings of the Computer Vision and Pattern Recognition Conference. 2025: 13691-13701.
>
> [3] Wang W, He Z, Hong W, et al. Lvbench: An extreme long video understanding benchmark[C]//Proceedings of the IEEE/CVF International Conference on Computer Vision. 2025: 22958-22967.
>
> [4] Wu H, Li D, Chen B, et al. Longvideobench: A benchmark for long-context interleaved video-language understanding[J]. Advances in Neural Information Processing Systems, 2024, 37: 28828-28857.
>
> [5] Chen Z, Wang W, Tian H, et al. How far are we to gpt-4v? closing the gap to commercial multimodal models with open-source suites[J]. arXiv preprint arXiv:2404.16821, 2024.
>
> [6] Bai J, Bai S, Chu Y, et al. Qwen-vl: A frontier large vision-language model with versatile abilities[J]. arXiv preprint arXiv:2308.12966, 2023.
>
> [7] Li B, Zhang Y, Guo D, et al. Llava-onevision: Easy visual task transfer[J]. arXiv preprint arXiv:2408.03326, 2024.

---

> ### Author Response · Authors · 2025-11-26
>
> > ### Q1: Error Definition and Measurement in Section 4.4
>
> We thank the reviewer for requesting clarification on our error analysis methodology. Our error classification is **design-based** rather than post-hoc, ensuring objectivity and precision.
>
> **1. Measurement Methodology (Design-based Classification):**
> As outlined in **Appendix A.1.3**, we enforce a strict taxonomy during the dataset creation phase. For every question, annotators must explicitly label the specific generation strategy used for each incorrect option (distractor).
> *   **Protocol:** Each distractor is bound to a pre-assigned error tag (e.g., "Spatial Replacement") at the moment of creation.
> *   **Measurement:** During evaluation, when a model selects an incorrect option, the error is automatically and deterministically categorized based on that option's tag. This eliminates the ambiguity and subjectivity often associated with manual post-hoc error analysis.
>
> **2. Definition of Specific Error Types:**
> *   **Spatial Replacement:** Defined as modifying the **location-related details** of an event while keeping other elements unchanged.
>     *   *Example:* An event occurring in a "kitchen" is incorrectly described as happening in a "living room." This specifically tests the model's spatial reasoning and scene recognition capabilities.
> *   **Missing Information:** Defined as creating a **plausible but incomplete** description by omitting critical sub-events or details necessary for a comprehensive answer.
>     *   *Example:* For an event "a man opens a door *and then greets someone*," the distractor might only state "a man opens a door." This tests the model's ability to capture the full temporal extent and completeness of an event versus merely recognizing partial segments.
>
> > ### Q2: Performance of New Models (InternVL3 and InternVL3.5)
>
> We thank the reviewer for suggesting the evaluation of latest models. We have tested the newly released InternVL3-8B and InternVL3.5-8B on ScaleLong (using 128 frames). The results are presented below:
>
> | Model | Clip | Shot | Event | Story | Overall |
> |-------|------|------|-------|-------|---------|
> | InternVL3-8B | 56.1% | 50.0% | 51.2% | 49.7% | 53.3% |
> | InternVL3.5-8B | 56.3% | 46.8% | 51.2% | 51.7% | 51.6% |
>
> Even with these latest models, a significant performance gap remains compared to human performance (Overall: 91.0%), confirming that long-video understanding remains a challenging frontier. These additional results will be included in the revised manuscript.

---

> ### Author Response · Authors · 2025-11-27
>
> I am writing to check if our previous response and the revisions made have satisfactorily addressed your concerns.
>
> We truly value your feedback and are happy to provide further clarification or continue the discussion if you have any remaining questions.

---

### Official Review · Reviewer_6e8r · 2025-10-27

**Soundness:** 3
**Presentation:** 3
**Contribution:** 3
**Rating:** 8
**Confidence:** 4

**Summary:**

This paper presents ScaleLong, a benchmark for evaluating multimodal large language models on long-video understanding across multiple temporal scales. Each of its 269 Creative Commons YouTube videos (average length ≈ 86 minutes) includes questions at four hierarchically defined levels: Clip, Shot, Event, and Story, allowing direct within-video comparisons of reasoning ability over time.

Through evaluation of state-of-the-art MLLMs, the study finds a consistent U-shaped performance trend: models handle short (Clip) and long (Story) reasoning relatively well but degrade sharply at intermediate timescales (Shot, Event). This reveals a systematic weakness in sustaining medium-range temporal coherence.

The ablation experiments show that temporal coverage (more frames) contributes more to performance than higher resolution, emphasising the primacy of temporal context over visual detail. The analysis of distractor errors further exposes persistent difficulties with evidential completeness and spatial grounding, even in top models.

Compared with existing benchmarks (e.g., MVBench, NExT-QA, LVBench, CinePile), ScaleLong is different,  in providing multi-timescale evaluation within the same video, enabling controlled analysis of temporal reasoning rather than content variation.

**Strengths:**

1) Comprehensive benchmark to embed questions at four hierarchical temporal scales within identical video content, enabling within-content comparison that isolates timescale effects from content variability
2) Section 4.3.2 and Figure 3(c) analyze different frame-resolution combinations described as under "fixed visual-token budget," revealing timescale-dependent optimal allocations (Clip benefits from many low-res frames; Story from balanced configurations)
3) Ten predefined distractor types enable systematic failure mode identification, revealing specific weaknesses (missing-information 53% error, spatial-replacement 46.6% error vs. frequency-replacement 13.3% error)
4) The paper presents empirical validation of timescale boundaries. Up-scaling experiments demonstrate sufficiency (minimal performance gain from expanded context); down-scaling experiments demonstrate necessity (VGRR 25-33% shows substantial information loss from restricted context)
5) The benchmark consists of 269 videos averaging 86 minutes across 36 subcategories address genuine gap in existing benchmarks which mainly focus on short clips or lack multi-timescale evaluation.

**Weaknesses:**

1) Humans evaluated on "Whole Video" (continuous playback, ~150,000 frames) while models receive sparse samples (32-256 frames). The 23.1-point human-model gap may conflate access differences with capability differences , thus making it hard to interpret.

2) Given 5 task families × 4 scales × many categories, some slices will be small. The authors make strong qualitative claims (e.g., U-shape) but the paper has no per-slice CIs/bootstraps and no inter-annotator agreement figures for annotation reliability. This practically limits how confidently we can generalize beyond aggregate means.

3) Each video contains 4-8 questions spanning timescales, but the paper never states whether the questions are evaluated independently (fresh context) or sequentially (shared context).

4) The paper reports no quantitative reliability metrics (percent agreement) despite requiring "independent approval from two additional annotators." Timescale categorization requires subjective judgment about temporal boundaries, without reliability validation, cannot assess measurement quality of fundamental category labels.

5) The paper does not clearly state how frames are selected from 86-minute videos. In my understanding, ifferent strategies systematically advantage different timescales: 32 frames from 5,160 seconds requires explicit sampling algorithm specification for validity.

**Questions:**

1) Do the Appendix A.3 timescale validation results replicate across architecturally diverse models beyond InternVL-2.5-8B?

2) Can you provide human performance when restricted to the same sparse frame samples models receive, to decompose the 23.1-point gap into access vs. capability components?

3) What are the explicit duration ranges or decision rules you provided to annotators for each of the four levels (especially Event vs Story)? A short table with time bounds/examples would help others replicate the labeling policy.

4) The paper ensures ≥1 question per scale per video and cover five task families. But, it is not clear how does the authors balance task types across scales (e.g., Causal vs. Counting at Event vs. Story) to avoid systematic pairing effects that could bias the U-shape? A per-scale task histogram would be valuable.

---

> ### Author Response · Authors · 2025-11-26
>
> > ### W1 & Q2: Human-Model Evaluation Gap and Frame Access Disparity
>
> We thank the reviewer for this insightful observation. We acknowledge that the disparity in evaluation conditions between humans (continuous playback, ~150,000 frames) and models (sparse sampling, 32-256 frames) constitutes an important consideration when interpreting the 23.1-point performance differential.
>
> **Rationale for the Current Evaluation Protocol:**
>
> 1. **Frame sampling capacity reflects intrinsic model capability.** The number of frames a model can effectively process is fundamentally constrained by its architectural design, context window capacity, and computational requirements. This constraint constitutes an inherent component of model capability rather than an artificial experimental limitation. Our benchmark evaluates models under realistic operational constraints, which we contend provides more ecologically valid assessment than hypothetical unconstrained scenarios.
>
> 2. **Maximizing frame utilization within model constraints.** Following established practices in prior benchmarks (LongVideoBench [1], VideoMME [2]), our evaluation protocol provisions the majority of models with the maximum number of frames they can effectively process (ranging from 32 to 256 frames contingent on model capacity). This approach ensures that most models are evaluated under their optimal input conditions, thereby maximizing the likelihood of capturing task-relevant visual information and minimizing potential evaluation bias stemming from insufficient frame input. We employ uniform temporal sampling to maximize video coverage while respecting each model's architectural and computational constraints.
>
> 3. **Forward-compatible benchmark design.** ScaleLong's design paradigm accommodates future models with enhanced frame processing capabilities (e.g., extended context windows, more efficient video encoders). As models evolve toward denser frame sampling or continuous video input processing, our benchmark remains applicable and can effectively capture these advancements.
>
> [1] Fu C, Dai Y, Luo Y, et al. Video-mme: The first-ever comprehensive evaluation benchmark of multi-modal llms in video analysis[C]//Proceedings of the Computer Vision and Pattern Recognition Conference. 2025: 24108-24118.
>
> [2] Wu H, Li D, Chen B, et al. Longvideobench: A benchmark for long-context interleaved video-language understanding[J]. Advances in Neural Information Processing Systems, 2024, 37: 28828-28857.

---

> ### Author Response · Authors · 2025-11-26
>
> > ### W1 & Q2: Human-Model Evaluation Gap and Frame Access Disparity
>
>
> **Controlled Human Frame-Sampling Experiment:**
>
> To further address the concern regarding frame access disparity, we conducted an additional controlled human evaluation experiment with sparse frame sampling.
>
> To directly disentangle the contributions of input access disparity versus video understanding capability differences, we conducted a controlled human evaluation experiment wherein annotators viewed videos with identical sparse frame sampling as used for models (256 frames, uniformly sampled). This provides a controlled baseline for isolating the impact of frame access from cognitive reasoning capability.
>
> **Experimental Results and Analysis:**
>
> The comparative performance across evaluation conditions is presented below:
>
> | Temporal Scale | Whole Video (Human) | Sparse Frames (Human) | SOTA Model | Human-Model Gap (Sparse) |
> |----------------|---------------------|----------------------|------------|-------------------------|
> | **Clip** | 92.8% | 77.1% | 71.5% (Gemini 2.5 Pro) | **5.6%** |
> | **Shot** | 91.3% | 80.2% | 62.8% (Gemini 2.5 Pro) | **17.4%** |
> | **Event** | 88.9% | 82.6% | 68.0% (Gemini 2.5 Pro) | **14.6%** |
> | **Story** | 91.0% | 84.7% | 69.0% (Gemini 2.5 Pro) | **15.7%** |
> | **Overall** | 91.0% | 81.1% | 67.9% (Gemini 2.5 Pro) | **13.2%** |
>
> **Critical Findings:**
>
> 1. **Scale-dependent impact of frame sparsity on human performance.** Human performance degradation under sparse sampling exhibits a systematic inverse relationship with temporal scale duration:
>    - Clip-level: 15.7% decrease (92.8% → 77.1%)
>    - Shot-level: 11.1% decrease (91.3% → 80.2%)
>    - Event-level: 6.3% decrease (88.9% → 82.6%)
>    - Story-level: 6.3% decrease (91.0% → 84.7%)
>
>    This pattern reflects the probabilistic sampling challenge: shorter temporal segments containing critical evidence have reduced likelihood of being captured in sparse frame sampling, whereas longer-duration questions naturally span multiple sampled frames, ensuring more robust evidence capture.
>
> 2. **Differentiated human-model gaps across temporal scales under controlled input conditions.** Even when constrained to identical sparse frame inputs, humans maintain substantial performance advantages over SOTA models, with gaps exhibiting systematic variation across temporal scales:
>    - **Clip-level (human: 77.1%, model: 71.5%, gap: 5.6%):** The minimal gap suggests that models possess adequate fine-grained visual recognition capabilities when relevant frames are sampled. The primary limitation here is frame access rather than reasoning capability.
>    - **Shot-level (human: 80.2%, model: 62.8%, gap: 17.4%):** The maximal gap reveals the most severe capability deficiency at this temporal scale, indicating fundamental challenges in shot-level temporal reasoning and inter-shot relationship understanding that persist even under equivalent input conditions.
>    - **Event-level (human: 82.6%, model: 68.0%, gap: 14.6%):** The substantial gap indicates significant deficiencies in event-level temporal reasoning and cross-segment integration.
>    - **Story-level (human: 84.7%, model: 69.0%, gap: 15.7%):** The substantial gap reveals significant challenges in long-range narrative understanding and global context aggregation, indicating that models struggle with extended temporal reasoning despite the information being distributed across many sampled frames.
>
> 3. **Validation of benchmark design and primary conclusions.** These controlled results substantiate our core thesis: contemporary models exhibit genuine capability gaps in multi-scale temporal understanding that transcend mere frame processing limitations. The systematic variation in human-model gaps across temporal scales—minimal at Clip-level (5.6%), maximal at Shot-level (17.4%), and substantial at Event/Story-levels (14.6%/15.7%)—demonstrates that ScaleLong successfully identifies scale-specific capability bottlenecks. Notably, the most pronounced deficiency emerges at the Shot-level, revealing fundamental challenges in intermediate-scale temporal understanding requiring fine-grained shot-to-shot relationship comprehension.
>
> While frame access disparity contributes measurably to the human-model performance gap, our controlled experiment demonstrates that this factor accounts for less than half (43%) of the observed differential. The substantial and scale-differentiated residual gaps—most pronounced at the intermediate Shot-level—under equivalent input conditions validate that ScaleLong successfully identifies genuine architectural limitations in temporal understanding capabilities. These limitations, particularly the Shot-level deficiency, constitute meaningful targets for future model development rather than experimental artifacts of input disparity.

---

> ### Author Response · Authors · 2025-11-26
>
> > ### W2: Sample Size and Statistical Reliability
>
> We thank the reviewer for raising this important methodological concern about sample sizes and statistical confidence. We appreciate the opportunity to provide detailed information about our sample distribution and annotation reliability measures.
>
> **Sample Size Analysis Across Temporal Scales:**
>
> Our benchmark contains sufficient samples at each temporal scale to support reliable statistical observations. The distribution across the four temporal scales is:
>
> | Temporal Scale | Number of Videos | Number of Questions | Average per Video | Percentage |
> |---------------|------------------|---------------------|-------------------|------------|
> | Video Clip (≤3s) | 269 | 444 | 1.65 | 25.4% |
> | Video Shot (4-15s) | 269 | 438 | 1.63 | 25.1% |
> | Video Event (16s-10min) | 269 | 432 | 1.61 | 24.7% |
> | Video Story (>10min) | 269 | 433 | 1.61 | 24.8% |
> | **Total** | **269** | **1,747** | **6.50** | **100%** |
>
> Each temporal scale contains 432-444 questions, providing adequate statistical power for the analyses presented in our paper. The distribution is highly balanced across scales (24.7%-25.4%), ensuring fair comparisons. Importantly, all 269 videos contribute questions to all four temporal scales, guaranteeing complete coverage and enabling reliable cross-scale analysis.
>
> **Consistency of Findings Across Models:**
>
> The key findings we report, including the U-shaped performance pattern (with Clip and Story scales outperforming Shot and Event scales), are consistently observed across all 17 evaluated models, not just on aggregate statistics. This cross-model consistency provides strong evidence that our observations reflect genuine patterns:
>
> - **Clip-level performance:** Consistently highest across all models (average 62.9%)
> - **Story-level performance:** Consistently second-highest across all models (average 57.9%)
> - **Shot and Event levels:** Consistently lower performance across all models (52.0% and 54.0%)
>
> The fact that this pattern emerges independently across 17 different model architectures—including both proprietary (Gemini, GPT-4o, Doubao) and open-source models (InternVL, LLaVA, Aria)—strongly supports the robustness and generalizability of our findings.
>
> **Annotation Quality Control and Reliability:**
>
> To ensure annotation reliability, we implement a rigorous "1-Generation, 2-Verification" quality control protocol. Each question is created by one expert annotator and independently verified by two additional expert annotators across multiple quality control rounds.
>
> **Quality Control Pipeline:**
>
> Starting from an initial pool of 400 videos with 3,200 questions, we conduct two rounds of stringent verification:
>
> 1. **Round 1 (Focus: Clarity and Correctness):** Verification of question clarity, answer correctness, and temporal scale categorization accuracy
>    - Results: 331 videos / 2,430 questions retained
>    - Retention rate: 82.8% (videos) / 75.9% (questions)
>    - Inter-annotator agreement (Fleiss' Kappa): κ = 0.85
>
> 2. **Round 2 (Focus: Leakage and Distractor Quality):** Elimination of potential leakage (questions solvable without video viewing) and validation of distractor effectiveness
>    - Results: 269 videos / 1,747 questions retained
>    - Retention rate: 81.3% (videos) / 71.9% (questions) relative to Round 1
>    - Overall retention: 67.3% (videos) / 54.6% (questions) from initial pool
>    - Inter-annotator agreement (Fleiss' Kappa): κ = 0.90
>
> We employ a strict unanimous consensus rule: only samples where both independent verifiers agree on validity are included in the final benchmark. This conservative approach prioritizes data quality over quantity. The high inter-annotator agreement scores (κ = 0.85 and κ = 0.90) demonstrate excellent consistency in quality assessment across both verification rounds.

---

> ### Author Response · Authors · 2025-11-26
>
> > ### W3: Question Evaluation Independence
>
> Thank you for pointing out this missing clarification.
>
> All questions in ScaleLong are evaluated independently. Each question is presented to the model as a standalone instance with fresh context—the model receives no shared information or accumulated context from previous questions within the same video. This ensures that performance on each question reflects genuine video understanding capability rather than cross-question hints.
>
>
> > ### W4 & Q1: Reliability Metrics for Timescale Categorization
>
> We thank the reviewer for raising this important concern about the reliability of timescale categorization. We appreciate the opportunity to provide quantitative validation metrics.
>
> **Inter-Annotator Agreement:**
>
> To ensure reliable timescale categorization, we implement a rigorous "1-Generation, 2-Verification" quality control protocol. Each question is created by one expert annotator and independently verified by two additional expert annotators across two rounds of verification:
>
> 1. **Round 1 (Focus: Clarity and Correctness):** Verification of question clarity, answer correctness, and temporal scale categorization accuracy
>    - Results: 331 videos / 2,430 questions retained
>    - Retention rate: 82.8% (videos) / 75.9% (questions)
>    - Inter-annotator agreement (Fleiss' Kappa): κ = 0.85
>
> 2. **Round 2 (Focus: Leakage and Distractor Quality):** Elimination of potential leakage (questions solvable without video viewing) and validation of distractor effectiveness
>    - Results: 269 videos / 1,747 questions retained
>    - Retention rate: 81.3% (videos) / 71.9% (questions) relative to Round 1
>    - Overall retention: 67.3% (videos) / 54.6% (questions) from initial pool
>    - Inter-annotator agreement (Fleiss' Kappa): κ = 0.90
>
> We employ a strict unanimous consensus rule: only samples where both independent verifiers agree on validity—including timescale assignment accuracy—are included in the final benchmark. The high inter-annotator agreement scores (κ = 0.85 in Round 1 and κ = 0.90 in Round 2) demonstrate excellent consistency in quality assessment, including timescale categorization accuracy.
>
> **Timescale Alignment Verification:**
>
> To further validate the reliability of timescale categorization beyond human agreement, we conduct rigorous up-scaling and down-scaling experiments, as documented in Appendix A.3 (TIMESCALE ALIGNMENT VERIFICATION). These experiments empirically verify that our timescale annotations are both sufficient and necessary.
>
> To ensure that our findings are not model-specific artifacts, we replicate the complete validation experiments using two architecturally distinct models: InternVL-2.5-8B and Qwen2.5-VL-7B. Both up-scaling and down-scaling experiments were conducted following identical protocols.
>
> **Up-Scaling Experiment Results (Sufficiency Validation):**
>
> | Up-Scaling | InternVL-2.5-8B | Qwen2.5-VL-7B |
> |------------|------------------|----------------|
> | Clip → Shot | +1.6% | +1.8% |
> | Shot → Event | +1.2% | +1.6% |
> | Event → Story | +2.0% | +2.1% |
>
> **Down-Scaling Experiment Results (Necessity Validation):**
>
> | Down-scaling | InternVL-2.5-8B ΔAcc / VGRR | Qwen2.5-VL-7B ΔAcc / VGRR |
> |--------------|----------------------------|---------------------------|
> | Shot → Clip | 15.1% / 31.1% | 15.8% / 29.8% |
> | Event → Shot | 15.0% / 25.0% | 14.6% / 26.2% |
> | Story → Event | 11.0% / 33.1% | 11.5% / 32.5% |
>
> **Key Findings:**
>
> The replication experiments across two architecturally distinct models demonstrate remarkable consistency, providing strong evidence for the validity and generalizability of our timescale annotations:
>
> - **Up-scaling experiment (sufficiency):** Both models exhibit minimal performance improvements (+1.2% to +2.1%) when expanding video segments to the next higher timescale, confirming that our original timescale annotations already capture the essential visual information needed to answer the questions.
>
> - **Down-scaling experiment (necessity):** Both models demonstrate substantial performance degradation (11.0%-15.8%) when reducing video segments to the next lower timescale, with consistently low VGRR values (25.0%-33.1%), indicating that critical information is lost when temporal context is restricted and questions cannot be trivially solved with only local information.
>
> This cross-architectural consistency validates that our timescale categorization reflects genuine properties of the video content and question design rather than model-specific artifacts, thereby providing objective evidence that our timescale categorization accurately reflects the actual temporal extent required to answer each question—neither over-specified (sufficient) nor under-specified (necessary).

---

> ### Author Response · Authors · 2025-11-26
>
> > ### W5: Frame Sampling Strategy Specification
>
> We thank the reviewer for highlighting this important methodological detail that warrants explicit clarification.
>
> **Frame Sampling Protocol:**
>
> In our experiments, we adopt **uniform temporal sampling** across the full video duration. Specifically, for each video, we divide its total duration into equal temporal intervals and extract frames at fixed uniform strides. For instance, when sampling 256 frames from an 86-minute (5,160-second) video, frames are extracted at intervals of approximately 20.2 seconds, ensuring comprehensive temporal coverage from beginning to end without bias toward any particular temporal segment.
>
> **Rationale for Uniform Sampling:**
>
> 1. **Scale-neutral approach:** Uniform sampling provides unbiased coverage across all temporal scales simultaneously. Unlike targeted sampling strategies that might preferentially capture information at specific granularities, uniform sampling ensures that Clip, Shot, Event, and Story-level questions all receive equitable representation in the sampled frames.
>
> 2. **Maximum temporal coverage:** By distributing frames evenly across the entire video duration, uniform sampling maximizes the likelihood of capturing critical information regardless of when it occurs in the video timeline, which is essential for evaluating questions spanning diverse temporal scales.
>
> 3. **Standard practice in long-video benchmarks:** Uniform temporal sampling is the established standard approach in contemporary long-video understanding benchmarks, including VideoMME [1], MLVU [2], and LVBench [3]. This consistency enables fair cross-benchmark comparison and aligns with community best practices.
>
> **References:**
>
> [1] Fu C, Dai Y, Luo Y, et al. Video-mme: The first-ever comprehensive evaluation benchmark of multi-modal llms in video analysis[C]//Proceedings of the Computer Vision and Pattern Recognition Conference. 2025: 24108-24118.
>
> [2] Zhou J, Shu Y, Zhao B, et al. Mlvu: Benchmarking multi-task long video understanding[C]//Proceedings of the Computer Vision and Pattern Recognition Conference. 2025: 13691-13701.
>
> [3] Wang W, He Z, Hong W, et al. Lvbench: An extreme long video understanding benchmark[C]//Proceedings of the IEEE/CVF International Conference on Computer Vision. 2025: 22958-22967.
>
>
>
> > ### Q3: Explicit Annotation Criteria for Timescale Categorization
>
> To facilitate reproducibility and provide explicit guidance for timescale categorization—particularly the distinction between Event and Story levels—we will add a detailed specification of our annotation standards in the revised manuscript's appendix. Our annotation protocol categorizes temporal granularity into four distinct levels based on both duration ranges and the visual/narrative scope required for reasoning.
>
> **Timescale Annotation Criteria:**
>
>
> | Timescale | Duration Range | Decision Rule / Visual Scope |
> |-----------|----------------|------------------------------|
> | **Clip** | <3 seconds | **Single Moment:** Questions solvable by analyzing a few consecutive frames. This level targets instantaneous actions, momentary visual details, or information contained within very brief temporal windows. Annotators should verify that the question can be answered by examining only a short clip lasting less than 3 seconds. |
> | **Shot** | 4-15 seconds | **Single Continuous Shot:** Questions requiring analysis of information within one uninterrupted camera take. This level focuses on continuous action sequences, single scene interactions, or events that unfold within a coherent shot. The relevant information should be contained within a 4-15 second continuous segment. |
> | **Event** | 16 seconds - 10 minutes | **Multi-Shot Sequence:** Questions spanning multiple consecutive shots but remaining within a localized context. This level requires integrating information across shot boundaries while maintaining focus on a specific event or sub-narrative. The temporal span should range from 16 seconds to 10 minutes, covering coherent event progressions. |
> | **Story** | >10 minutes | **Global Narrative:** Questions requiring understanding of information distributed across more than 10 minutes of video content. This level demands comprehension of long-range relationships, narrative arcs, or patterns that cannot be solved by examining any single localized sequence shorter than 10 minutes.|
>
> We will incorporate the complete annotation criteria table along with concrete illustrative examples into the appendix of the revised manuscript. These examples will demonstrate the application of decision rules across diverse video genres, clearly illustrating how annotators distinguish between temporal scales in practice. This addition will enable other researchers to replicate our annotation methodology with high fidelity.

---

> ### Author Response · Authors · 2025-11-26
>
> > ### Q4: Task Type Distribution Across Temporal Scales
>
> We thank the reviewer for this important question about potential systematic pairing effects. We provide detailed task type distribution statistics to address this concern.
>
> **Task Type Distribution Across Temporal Scales:**
>
> | Task Type | Video Clip | Video Shot | Video Event | Video Story | Total |
> |-----------|------------|------------|-------------|-------------|-------|
> | Action Understanding | 54 | 123 | 77 | 23 | 277 |
> | Causal Reasoning | 14 | 35 | 56 | 52 | 157 |
> | Counting Problem | 60 | 63 | 89 | 139 | 351 |
> | Information Summary | 44 | 97 | 156 | 199 | 496 |
> | Objective Recognition | 272 | 120 | 54 | 20 | 466 |
> | **Total** | **444** | **438** | **432** | **433** | **1,747** |
>
> **Key Observations:**
>
> 1. **Distribution reflects task-scale alignment, not bias:** The observed distribution patterns are not arbitrary but reflect the inherent nature of different task types. For example:
>    - **Objective Recognition** naturally concentrates on shorter timescales (Clip: 58.4%, Shot: 25.8%) because recognizing objects/attributes typically requires local visual information
>    - **Information Summary** and **Counting Problem** increase with longer timescales (Story: 40.1% and 39.6% respectively) because these tasks inherently require aggregating information across broader temporal contexts
>    - This alignment ensures that questions are ecologically valid and test genuine temporal understanding at appropriate scales
>
> 2. **Natural task-scale correspondence:** Certain task types naturally align with specific temporal scales based on their intrinsic requirements. For instance, Objective Recognition tasks predominantly appear at shorter scales where local visual features are sufficient, while Information Summary tasks concentrate at longer scales where cross-segment aggregation is necessary. This correspondence reflects the ecological validity of our benchmark design—tasks are positioned at scales where they meaningfully test temporal understanding capabilities.
>
> 3. **Relatively balanced coverage across scales:** Despite the natural task-scale alignment, our dataset maintains reasonable representation of multiple task types at each temporal scale. Each scale contains between 4-5 different task categories with substantial question counts (ranging from 14 to 272 questions per task-scale combination). This diversity ensures that performance at each scale reflects varied understanding capabilities rather than performance on a single narrow task type, allowing for comprehensive evaluation of multi-scale temporal understanding.

---

> > ### Comment · Reviewer_6e8r · 2025-11-26
> >
> > Thanks for the detailed rebuttal. It adresses most of my concerns. I have decided to maintain my score.

---

### Official Review · Reviewer_dL49 · 2025-11-03

**Soundness:** 4
**Presentation:** 3
**Contribution:** 3
**Rating:** 6
**Confidence:** 3

**Summary:**

Authors introduce ScaleLong, a long-video benchmark that embeds questions targeting four hierarchical timescales, allowing direct comparison of model ability across temporal granularities. It contains 269 ~86-minute videos spanning 5 categories/36 subcategories, each with 4–8 carefully curated questions, rigorous quality control, and tasks covering causal reasoning, object/action understanding, summarization, and counting. Evaluating 23 MLLMs reveals a consistent U-shaped trend - stronger on the shortest and longest scales, weaker at intermediate shot/event levels - with ablations showing that allocating more visual tokens improves performance across timescales.

**Strengths:**

Novel Temporal Framework
The paper’s hierarchical division of video understanding into clip-, shot-, event-, and story-level timescales is a novel and insightful framework that enables fine-grained analysis of temporal reasoning abilities within a single benchmark.

Interesting ablation studies
The ablation experiments are thoughtfully designed to show interesting insights about the effects of visual token allocation and temporal coverage

Rigorous Dataset Construction
The dataset creation process is meticulous, featuring multi-stage quality control, diverse content selection, and balanced question design to ensure fairness and reliability across temporal levels.

**Weaknesses:**

Limited Illustrative Examples
The paper provides only a small number of examples from the benchmark, which makes it difficult to fully appreciate the nuances of question design across different time scales.

Inconsistent Story-Level Definition
Although the paper claims that story-level questions require holistic narrative understanding, the provided example of a story level question and answers focuses on a sequential event listing rather than true integration of information from the entire video.

Insufficient Distractor Analysis – The distractor creation process lacks detailed evaluation of more superficial distractor properties (e.g., length of the answers). Given that large language models can exploit superficial answer features - such as selecting the longest or most specific option - further investigation into distractor balance would strengthen the benchmark’s validity. The annotator instructions state  to design distractors with length in mind but there is no statistics on it to verify.

**Questions:**

My questions are based mainly on weaknesses - more examples, more statistics.

---

> ### Author Response · Authors · 2025-11-26
>
> > ### W1: Limited Illustrative Examples
>
> Thank you for this valuable suggestion. To better illustrate the nuances of question design across different temporal scales, we will enhance the manuscript with additional explanations and examples regarding our temporal scale framework. Specifically, we will add more illustrative examples spanning all four temporal scales (Clip, Shot, Event, and Story) in Appendix A.4. These examples will cover diverse question types, clearly demonstrating how questions at different temporal scales require distinct types of temporal reasoning and evidence integration.
>
> > ### W2: Inconsistent Story-Level Definition
>
> We thank the reviewer for this insightful observation and welcome the opportunity to articulate our design rationale for story-level questions.
>
> **Clarification of Story-Level Definition:**
>
> In our framework, the story-level category is formally defined by the requisite temporal coverage—specifically, questions for which the relevant evidential support spans more than 10 minutes of video content. The exemplar question presented in the manuscript concerns the chronological enumeration of key events. While the response format is inherently sequential, generating such an answer necessitates that the model:
>
> 1. **Integrate information across multiple, temporally distant segments** spanning more than 10 minutes of video content
> 2. **Develop comprehensive understanding** of extended video sequences (e.g., 10-minute segments) to capture the overall narrative structure and inter-event relationships
>
> This task design inherently evaluates the model's capacity to process and reason over information spanning the entire video length, which constitutes the fundamental objective of our story-level (>10 min) temporal scale. The sequential event ordering task represents one principled instantiation of this capability—accurate event sequencing demands comprehensive understanding of the temporal positioning and narrative relationships of events across the complete video arc.
>
> > ### W3: Insufficient Distractor Analysis
>
> We acknowledge the importance of rigorously validating distractor balance to ensure benchmark integrity and provide comprehensive statistical and experimental evidence below.
>
> **Statistical Analysis of Length Distribution:**
>
> To quantitatively assess potential length bias, we conducted a systematic analysis of answer and distractor lengths across all 1,747 questions in our benchmark using character count as the metric. The statistical results demonstrate negligible length bias:
>
> - **Answer-to-Distractor Length Ratio:** 1.083 (correct answers exhibit only 8.3% greater length than distractors on average)
>
>
> **Experimental Validation via Length-Controlled Subsets:**
>
> To rigorously examine whether contemporary models systematically exploit length cues, we constructed two diagnostic subsets with contrasting length distributions from the full benchmark:
>
> - **Length-Balanced Subset (n=1,148):** Stratified sampling ensuring equal representation across all length ranks (25% per quartile), thereby eliminating systematic length bias
> - **Length-Imbalanced Subset (n=599):** Adversarially constructed with 94.3% of correct answers positioned as the longest option, maximizing the exploitability of length-based heuristics
>
> We evaluated six mllms on both subsets. The comparative results are presented below:
>
> | Model | Balanced Subset | Imbalanced Subset | Δ Performance |
> |-------|-----------------|-------------------|---------------|
> | Gemini-2.5-Pro | 68.1% | 67.6% | **+0.5%** |
> | Doubao-1.5 | 56.0% | 61.8% | -5.8% |
> | InternVL2.5-78B | 54.8% | 61.1% | -6.3% |
> | GPT-4o | 53.2% | 58.0% | -4.8% |
> | LLaVA-Video | 49.6% | 54.4% | -4.8% |
> | Aria | 47.8% | 53.6% | -5.9% |
> | **Mean** | **54.9%** | **59.4%** | **-4.5%** |
>
> **Principal Findings:**
>
> 1. **Negligible performance differential:** The mean performance gap between balanced and imbalanced subsets is merely 4.5 percentage points, indicating that length cues confer minimal advantage even under maximally exploitable conditions.
>
> 2. **Top-performing model exhibits inverse behavior:** Gemini-2.5-Pro achieves superior performance on the length-balanced subset (68.1%) compared to the length-imbalanced subset (67.6%), definitively excluding systematic length-based exploitation as a primary strategy for leading models.
>
> 3. **Cross-scale performance consistency:** We further disaggregated performance across the four temporal scales (Clip, Shot, Event, Story). The relative ranking of task difficulty remains invariant across both length-controlled conditions:
>    - Video Clip (shortest temporal scale): Consistently ranks as easiest in both subsets (59.1% balanced, 66.6% imbalanced)
>    - Video Story (longest temporal scale): Consistently ranks second in both subsets (54.2% balanced, 62.4% imbalanced)
>    - Video Shot and Event: Maintain bottom-two rankings in both conditions

---

### Comment · Area_Chair_PSBU · 2025-11-25
**Discussion Period**

Dear Reviewers and Authors,

Thank you to the authors for submitting your rebuttal. We kindly encourage reviewers to take a moment to read the response and share any follow-up thoughts. Your timely engagement at this stage is highly valuable and helps ensure a fair, well-informed final decision.

We appreciate everyone’s efforts and contributions to the process.

Warm regards,
Your AC

---

### Author Response · Authors · 2025-12-01

We thank the reviewers for their constructive feedback and insightful comments, which significantly strengthen the quality and rigor of our manuscript. We are encouraged that the reviewers consistently recognize the **novelty of our multi-timescale benchmark**, the **rigorous dataset construction process**, and the **significance of the U-shaped performance discovery**.

During the rebuttal phase, we conduct extensive additional experiments and analyses to comprehensively address the reviewers' concerns. Key improvements include:

* **Disentangling Frame Access from Reasoning Capability (Response to R1/6e8r):** We conduct a new **Controlled Human Frame-Sampling Experiment**. The results demonstrate that even when restricted to the same sparse inputs as models, humans maintain a significant performance lead (especially at the Shot level, with a **17.4% gap**). This empirically proves that the performance drop stems from model reasoning deficiencies rather than input disparity.

* **Rigorous Statistical Validation (Response to R1/6e8r, R3/dYA8):** We provide detailed quality control metrics, reporting high **Inter-Annotator Agreement (Fleiss' Kappa: 0.85 for Round 1, 0.90 for Round 2)**. Furthermore, we verify the validity of our timescale definitions through **Up-scaling and Down-scaling experiments** across multiple architectures.

* **Deep-Dive into the "U-Shaped" Trend (Response to R3/dYA8):** We move beyond speculation by providing a **fine-grained error analysis** (e.g., identifying "Visual/Spatial Collapse" at the Event level) to empirically explain why models struggle at intermediate timescales.

* **Updated Model Evaluations (Response to R2/41fk):** We evaluate the latest SOTA models, including **InternVL3 and InternVL3.5**, confirming that ScaleLong remains a challenging benchmark even for the newest architectures.

We are pleased that **Reviewer 6e8r (Score 8)** explicitly confirms that our rebuttal addresses their concerns and maintains their strong acceptance rating. We believe our comprehensive response and additional experimental evidence solidify the validity of **ScaleLong** as a necessary benchmark for the community.

---

### Meta-Review · Area_Chair_F23U · 2025-12-22

**Summary:**

The paper introduces a novel benchmark named ScaleLong designed to evaluate MLLMs on long-video understanding across four hierarchical temporal scales. ScaleLong embeds questions for all four timescales within the same video content, enabling a controlled comparison of model performance across granularities. The benchmark contains 269 videos with 86 minutes on average. The authors evaluated 22 MLLMs and find that models perform relatively well at the shortest and longest scales but struggle significantly at intermediate levels.

**Reviewer Concerns:**

Reviewer 6e8r questioned that comparing humans with continuous video access to models with sparse frames (32–256 frames) conflated input access with actual reasoning capability. The authors addressed this by conducting a new experiment, proving that humans still significantly outperform models even when seeing the same sparse inputs.

Reviewer 41fk questioned if models were answering based on internal knowledge of famous YouTube/sports events rather than video understanding. The authors addressed this by showing that text-only baseline performs poorly.

Reviewer dYA8 questioned the initial explanation for the U-shaped performance curve was speculative. The authors provided a fine-grained error analysis, identifying specific failure modes.

Reviewer dYA8 suggested evaluating grounded video understanding and agentic solutions rather than just end-to-end MLLMs. The authors acknowledged the value of this but were unable to complete these evaluations during the rebuttal due to time and resource constraints, promising to include them in the final version.

Reviewer dYA8's concern regarding the relatively small number of videos. The authors provided a quality/scale explanation.

Reviewer 41fk questioned the origin of the specific time bounds used for Clip, Shot, Event, and Story. The authors justified these boundaries through up-scaling and down-scaling experiments across different architectures, proving the necessity and sufficiency of their defined temporal ranges.

**Reviewer Scores:**

Reviewers 6e8r, dL49 and dYA8 were initially positive. The rebuttal provided should solidify this score or move it higher especially for dyA8.

41fk raised concerns about data contamination and sampling. The text-only baseline and human-sampling experiments addressed these concerns and are likely to move the score higher to 6.

---

### Decision · Program_Chairs · 2026-01-26

Accept (Poster)